# Spatial-frequency channels, shape bias, and adversarial robustness

**Ajay Subramanian**
New York University
as15003@nyu.edu

**Elena Sizikova**
New York University
es5223@nyu.edu

**Najib J. Majaj**
New York University
najib.majaj@nyu.edu

**Denis G. Pelli**
New York University
denis.pelli@nyu.edu

## Abstract

What spatial frequency information do humans and neural networks use to recognize objects? In neuroscience, *critical band masking* is an established tool that can reveal the frequency-selective filters used for object recognition. Critical band masking measures the sensitivity of recognition performance to noise added at each spatial frequency. Existing critical band masking studies show that humans recognize periodic patterns (gratings) and letters by means of a spatial-frequency filter (or "channel") that has a frequency bandwidth of one octave (doubling of frequency). Here, we introduce critical band masking as a task for network-human comparison and test 14 humans and 76 neural networks on 16-way ImageNet categorization in the presence of narrowband noise. We find that humans recognize objects in natural images using the same one-octave-wide channel that they use for letters and gratings, making it a canonical feature of human object recognition. Unlike humans, the neural network channel is very broad, 2-4 times wider than the human channel. This means that the network channel extends to frequencies higher and lower than those that humans are sensitive to. Thus, noise at those frequencies will impair network performance and spare human performance. Adversarial and augmented-image training are commonly used to increase network robustness and shape bias. Does this training align network and human object recognition channels? Three network channel properties (bandwidth, center frequency, peak noise sensitivity) correlate strongly with shape bias (51% variance explained) and robustness of adversarially-trained networks (66% variance explained). Adversarial training increases robustness but expands the channel bandwidth even further beyond the human bandwidth. Thus, critical band masking reveals that the network channel is more than twice as wide as the human channel, and that adversarial training only makes it worse. Networks with narrower channels might be more robust.

## 1 Introduction

The world can be visually perceived at many possible resolutions. It contains large and tiny objects, with coarse and fine features, so viewing it at different resolutions amplifies different features. Everyday human activities like reading, driving, and social interaction rely heavily on our ability to recognize objects. While doing so, what resolutions do we "see" the world in? Answering this question will help us understand what features are useful for object recognition.

37th Conference on Neural Information Processing Systems (NeurIPS 2023).

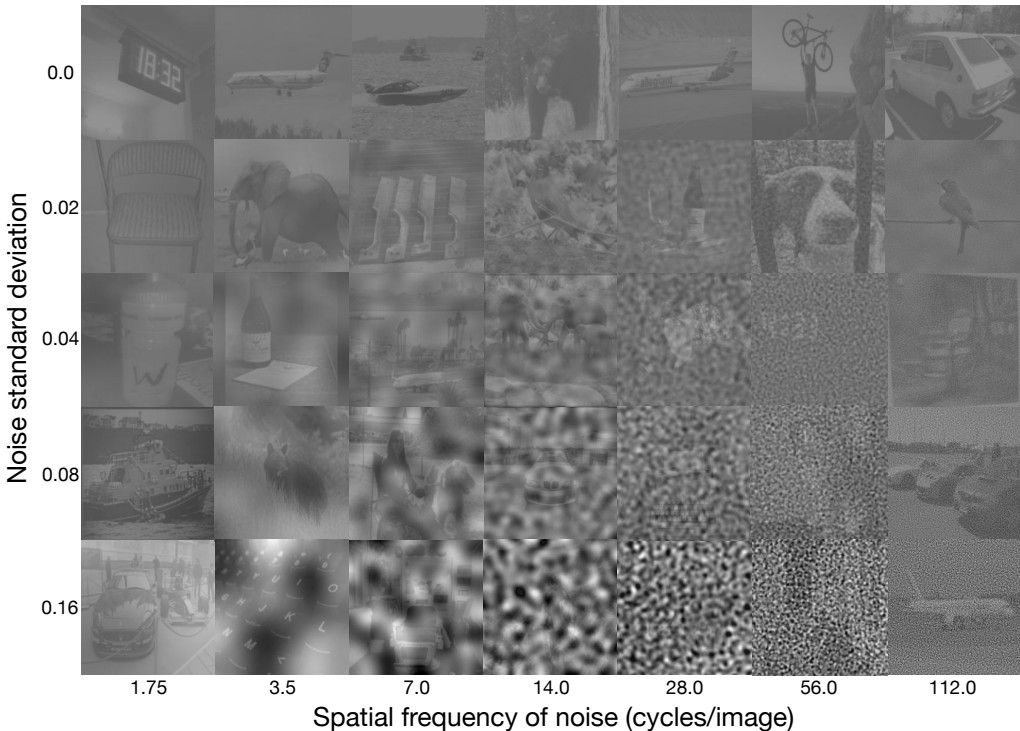

Figure 1: *Demo: Critical band masking reveals the spatial frequency channel used for object recognition.* Each cell in the grid contains a sample grayscale, contrast-reduced (20%) image from ImageNet [5], perturbed with Gaussian noise of standard deviation $\sigma \in \{0, 0.02, 0.04, 0.08, 0.16\}$ and filtered within one-octave-wide (doubling of frequency) spatial frequency bands centered at $f_{center} \in \{1.75, 3.5, 7.0, 14.0, 28.0, 56.0, 112.0\}$ cycles/image. Standard deviation of added noise increases from top to bottom, and spatial frequency increases from left to right (with constant power). Measure how far down each column you can go before being unable to recognize the object in the image. These *thresholds* measured across columns should trace out an inverted-U-shaped curve, typically centered at 28-56 cycles/image. This curve describes how much noise added at each spatial frequency limits your recognition performance, or in other words, what spatial frequency information in the image you most rely on for recognition, i.e., your "channel".

Formally, this question has been studied by examining how the brain filters visual stimuli. The visual system detects periodic patterns or gratings by means of parallel visual filters, each tuned to a band of spatial frequency [1]. Experiments by Solomon & Pelli [2], and later by Majaj et al. [3], showed that letter recognition across a wide-range of stimulus conditions (alphabets, fonts, sizes, low- and high-pass noise) is mediated by the same single, narrow (1-octave-wide) visual filter, which they called a "channel".

However, unlike letters and gratings, the real-world contains objects and features of various sizes, so we would expect people to make use of a wider band of spatial frequency in their recognition. Using a critical band masking task, first developed by Fletcher to identify auditory channels [4], and later used by Solomon & Pelli to find letter-recognition channels [2], we find that surprisingly, this is not the case. People use the same channel while performing 16-way ImageNet recognition [5], making it a canonical feature of human object recognition.

This raises the question, what spatial frequencies do deep-neural-network-based object recognition systems use, and how do they compare to the human channel? Given that these networks are infamously susceptible to high-frequency adversarial perturbations, we hypothesize that their channel must be unlike the human channel. Testing neural-network-based recognizers with the same 16-way ImageNet recognition task as above, we find that this is true, across various architectures and training strategies. The neural network channel in the 76 networks we tested is at least twice as wide as the human channel.

We then ask if this large, systematic difference between humans and networks could explain two important and well-documented points of divergence in their behavior: adversarial robustness and texture-vs-shape bias. It is well known in the machine learning literature that computer vision systems are highly-susceptible to targeted noise, known as adversarial perturbation, that is often imperceptible to human observers [6]. Additionally, convolutional networks primarily rely on texture information when recognizing images [7–9] whereas transformer-based networks and humans rely on shape [9, 10]. We show here that the spatial frequency channel provides a helpful way of looking at these issues. Wang et al. [11] found that adversarial perturbations generally contain most of their power at high spatial frequencies. Moreover, Lieber at al. [12] showed that people use higher spatial frequencies while discriminating texture patches. We find that properties of the network channel correlate strongly with shape bias and the robustness of adversarially-trained networks, albeit in the wrong direction — adversarial training makes the networks' frequency-selectivity less human-like.

To summarize our contributions:

1. We introduce critical band masking as a task for network-human comparison. We present data from 14 humans and 76 neural networks performing 16-way ImageNet categorization in the presence of narrowband noise.

2. Using our data, we characterize, for the first time, the spatial frequency channel used for natural object recognition by humans and networks. Humans use the same narrow, one-octave-wide channel for objects, letters, and gratings [2, 3]. Thus, this channel is a canonical feature of human object recognition. The neural network channel, across architectures and training procedures is more than twice as wide as the human channel.

3. We show that three channel properties (bandwidth, center frequency, peak noise sensitivity) correlate strongly with shape bias and robustness of adversarially-trained networks. Thus, these well-documented human-vs-network differences may be rooted in differences between their spatial-frequency channel. Adversarial training increases robustness at the cost of widening the already-too-wide network channel.

Our human and network datasets as well as code required to reproduce our experiments are available publicly at `https://github.com/ajaysub110/critical-band-masking`.

## 2 Related work

**Spatial-frequency channels.** The visual system detects periodic patterns or gratings by means of parallel visual filters, each tuned to a band of spatial frequency [1]. Critical band masking studies [4] revealed that the same single, narrow filter also mediates the recognition of letters [2, 3, 13], faces, and novel shapes [14]. Artificial neural networks also have frequency-based preferences. They are biased towards learning low-frequency functions [15] and prone to shortcut learning in both spatial and frequency domains [16, 17]. Existing work also suggests that robustness of a network is related to its spatial-frequency preferences [11, 18–21].

**Shape bias.** Humans are well known to rely mainly on shape features for lexical learning and object recognition tasks [10, 7]. ImageNet-pretrained convolutional networks, on the other hand, are biased towards texture [22, 23]. Even though ImageNet-pretrained transformers, like humans, are shape-biased [9], texture-vs-shape bias of networks is thought to be influenced mainly by training data and its augmentations rather than network architecture [8].

**Adversarial robustness.** Adversarial attacks are small perturbations that cause inputs to be misclassified [6, 24]. Although these perturbations are often imperceptible to humans, humans can in some cases decipher adversarial examples [25–27]. Recent work suggests that adversarial robustness of networks relates to their spatial frequency tuning [28, 18, 29].

**Comparing human and neural network vision.** The origins of deep learning are strongly tied to neuroscience [30] and the modern convolutional network architecture was inspired by properties of primate visual cortex [31]. More recently, there have been parallel efforts both to use neural networks to improve models of visual neuroscience [32–34] and to improve network robustness by taking inspiration from the human visual system [35]. These lines of work strongly rely on recent advancements in model-human comparison metrics which compare networks and humans across behavior [36, 37] and neural representations [34, 38].

# 3 Methods

## 3.1 Critical band masking

Solomon & Pelli developed a critical band masking task (based on Fletcher's auditory masking paradigm [4]) to find the human spatial-frequency channel for letter recognition [2]. We adapted their task to find the channel used by 14 human and 76 neural network observers to recognize objects in naturalistic images. The idea behind the task is as follows. Grayscale ImageNet images were converted to low-contrast (20%) and perturbed with Gaussian noise that was filtered within various spatial-frequency bands. Threshold contrasts for 50% categorization accuracy measured independently for each spatial-frequency condition trace out a curve that describes how sensitive human letter recognition is to noise at each spatial-frequency. Bands that have high noise-sensitivity are more useful for recognition because blocking them with noise severely impairs performance. In this way, critical band masking reveals the spatial-frequency channel useful for object recognition. Fig. 1 illustrates this idea using a demo.

**Images.** In all our experiments, human observers recognized objects in images from ImageNet [5], a popular dataset for neural network analysis. Since memorizing all 1000 categories of the original ImageNet is intractable for human observers, we used 16-class ImageNet [22], a subset which, using the WordNet hierarchy [39], is labeled according to 16 higher-level categories: airplane, bear, bicycle, bird, boat, bottle, car, cat, chair, clock, dog, elephant, keyboard, knife, oven, and truck. Due to constraints on experiment length, we used a fixed set of 1100 randomly sampled images for all experiments, that were roughly equally distributed across all 16 categories.

**Preprocessing.** Images were first resized to $256 \times 256$ and center-cropped to $224 \times 224$ (the standard protocol while handling ImageNet images) (Fig. 2A) before being converted to grayscale because noise, which we will add, is ill-defined for color images. Then, they were reduced to 20% of their original contrast to ensure that adding high-strength noise would not result in pixel-clipping at floor/ceiling values which would result in the effective added noise being non-Gaussian. Fig 2B1 shows sample images after these transformations.

**Noise generation and masking.** To the preprocessed images, we added Gaussian noise of 5 strengths (standard deviations) $(0.0, 0.02, 0.04, 0.08, 0.16)$, filtered into 7 octave-wide (doubling of frequency) spatial-frequency bands (centered at $1.75, 3.5, 7.0, 14.0, 28.0, 56.0, 128.0$ cycles/image) corresponding to the seven successive levels of a laplacian pyramid [40]. This gives us a total of 29 distinct noise conditions: 4 non-zero strengths $\times$ 7 spatial frequencies + 1 zero strength (Fig. 2B2). Both strength and spatial-frequency were sampled uniformly at random such that the 1100 images in our dataset were roughly equally distributed across both the noise conditions (combination of strength and spatial frequency; $\approx$31 images per condition) and categories ($\approx$68 images per category). Fig. 1 (smaller version in Fig. 2C) shows sample noise-masked images from our experiment for all 29 conditions as a demo. By following the instructions in the figure's caption, you should be able to see an inverted-U-shaped boundary between images you can and cannot recognize, peaking at 28-56 cycles/image, which represents your channel for object recognition.

## 3.2 Human psychophysics

14 human observers performed 16-way recognition of the noise-masked images for 1100 trials, divided into one training block of 50 trials and five test blocks of 210 trials. In each trial, the observer was first presented with a blank screen with a "+" at the center, which they were instructed to click to proceed. This ensured fixation at the center of the image at the start of presentation. Upon clicking, a randomly sampled image from our dataset was presented at the center of the screen (with a gray background) along with 16 buttons below it, representing the various categories (Fig. 2D). The observer was given unlimited time to identify the category of the image and click on the corresponding button. In training block trials, they would then see feedback on their response i.e., "correct" if they were right and "incorrect" along with the correct category if they were wrong. No feedback was given on test block trials (for screenshots of task, see Supplementary Material). We developed the study using LabJS [41], hosted it on JATOS [42], and recruited participants via Amazon Mechanical Turk. The experiment lasted roughly an hour and participants were paid $20 for their efforts. A standard IRB-approved (IRB-FY2016-404) consent form was signed by each observer, and demographic information was collected. All observers had either normal or corrected-to-normal vision. Data of observers who scored below 50% on the no-noise condition (0 strength, any spatial-frequency)

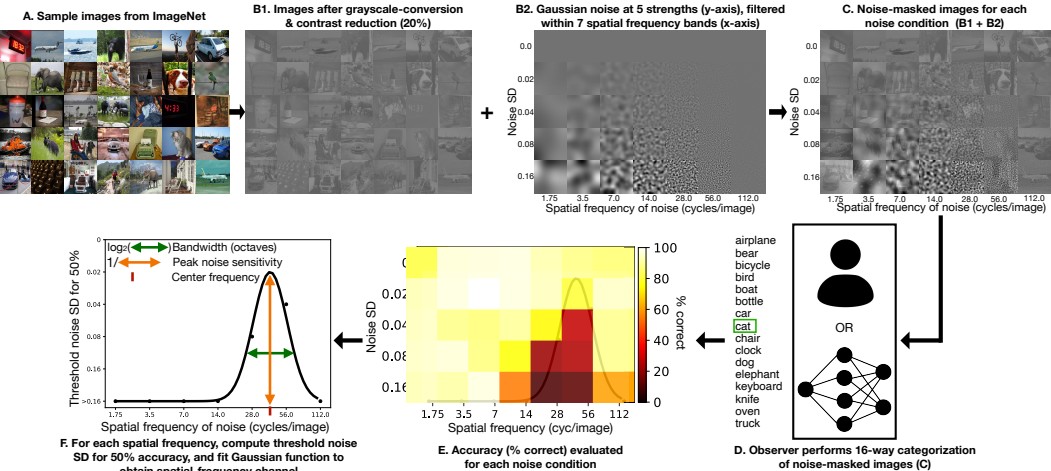

Figure 2: *Critical band masking stimuli, task and analysis*. **A.** 224 x 224 RGB images from ImageNet [5]. **B1.** Images after grayscale-conversion and contrast reduction to 20%. **B2.** $224 \times 224$ Gaussian noise of 5 standard deviations (SDs), bandpass-filtered within one-octave-wide spatial frequency (SF) bands centered at 7 values. This gives us a total of 29 distinct noise conditions (4 non-zero SDs $\times$ 7 SFs + 1 for 0 SD). **C.** Images in B1 + noise in B2 gives us sample noise-masked images for each noise condition. Our experiment used 34 different images for each condition. We show only one per condition here for the purpose of visualization. **D.** In our experiment, human and neural network observers attempt to categorize 1200 noise-masked images into 16 high-level object categories [22]. **E.** Heatmap showing % correct of a sample observer separately for each noise condition. **F.** For each spatial-frequency condition, threshold noise SD for 50% accuracy is computed (black points), and a Gaussian function is fit to obtain the observer's channel. This function is parameterized by center frequency (maroon), bandwidth in octaves (log full-width-half-max distance; green), and peak noise sensitivity (reciprocal of channel height at center frequency; orange). Noise filtered to conditions within the channel prevents recognition, and noise filtered to conditions outside the channel does not.

was discarded. Generally, researchers using online experiments recruit more than 100 participants. However, in our case, 14 was sufficient because we observed very small individual differences across participants (see Supplementary Material).

### 3.3 Neural network evaluation

We tested 76 pretrained neural networks on the same task that we used to test humans: 16-way categorization of noise-masked images from our critical band masking dataset (Fig. 2D). Networks and their pretrained weights were sourced primarily from PyTorch's torchvision library [43] and Geirhos et al.'s modelvshuman benchmark [36]. These networks spanned various architectures and training procedures (ImageNet-trained CNNs [43], self-supervised models [44–49], Big Transfer models [50], adversarially-trained models [51], vision transformers [52, 53], semi-weakly supervised models [54], Noisy Student [55], and VOneNet [35]) and hence, are a large, representative sample of the state of the art networks from deep learning literature. For a full list of the networks we tested, and the color coding used in figures, see Supplementary Material, and for network details, see [36]. We do not include the color codes in the main paper because with the exception of blue being used to represent adversarially-trained models, network-type is irrelevant to our analyses, i.e., our results apply to all networks unless specified. By default, all networks accepted $224 \times 224$ images, and output probabilities for all 1000 original Imagenet categories. Images were normalized to ImageNet pixel mean and variance before being input to the network, to respect training data statistics. Experiments run without normalization yielded very few networks (only 20 of 76) that performed above-threshold (>50% accuracy) even in the zero-noise condition. The probability of each coarse 16-class ImageNet category was computed as the average of the probabilities of the corresponding fine (1000-class ImageNet) categories (Supplementary Material of [22] justifies this choice). For each input image, the 16-class category with top-probability was selected as the network's response

and network accuracy (percent-correct) was calculated separately for each of the 29 noise conditions (strength, frequency combinations).

## 3.4 Metrics

Having collected data from humans and neural networks on our critical band masking task, we move to analysis. Accuracies (% correct for 16-way categorization) of each human observer and neural network were computed independently for each noise condition in our experiment. Fig. 2E summarizes this metric as a sample heatmap (in this case, for a sample human observer). Each cell in the heatmap represents a single noise condition (combination of standard deviation (SD) and spatial frequency; a cell in Figs. 1 and 2C), and its color, the categorization accuracy of the observer on that condition. For all analyses henceforth, we average across human participants because the cross-observer variance was very small (see Supplementary Material).

**Channel properties: center frequency, bandwidth, and peak noise sensitivity.** Next, we use the heatmap to estimate the observer's spatial frequency channel. The threshold noise strength (SD) for 50% accuracy was computed separately for each of the seven spatial frequency bands (column in the heatmap). These thresholds, depicted as black points in Fig. 2F, are a measurement of noise sensitivity i.e., how much noise of each spatial frequency has to be added for the observer's accuracy to drop to 50%. Fitting a Gaussian function then yields a curve that captures the noise sensitivity of recognition across spatial frequency bands. This curve is a suitable representation for the spatial frequency channel because it captures the importance of each spatial frequency band in the image for recognition by measuring how noise added to each band affects performance. The Gaussian function has been used extensively to describe spatial-frequency tuning in the brain [3] and also yields good fits across all human and network data (see Supplementary Material).

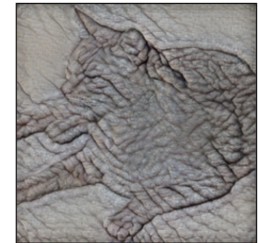

Figure 3: *A sample cue-conflict image from Geirhos et al.'s dataset that has a cat's shape and an elephant's texture. Source: [7].*

The fitting procedure is as follows. The threshold values computed earlier are first mapped to linear indices ($\{> 0.16, 0.16, 0.08, 0.04, 0.02\} \rightarrow \{0, 1, 2, 3, 4\}$). A Gaussian function $f(x) = Ae^{-\frac{(x-\mu)^2}{2\sigma^2}}$ having 3 parameters: peak height ($A$), mean ($\mu$), and standard deviation ($\sigma$) is then fit to the thresholds. These fitted parameters then used to calculate three properties that characterize the channel: bandwidth in octaves (log full-width at half-max = $2\sigma\sqrt{\ln 4}$), center frequency (frequency for peak noise sensitivity = $1.75 \times 2^{\mu}$), and peak noise sensitivity (scaled reciprocal of channel height = $2^{A-4}$). An octave is a doubling of frequency. Fig. 2F illustrates the rescaled Gaussian channel (black curve) and its three parameters (orange, maroon, green) for a sample observer, along with formulae to calculate the three channel properties.

**Shape bias.** In a later section, we study how the above-derived channel properties correlate with two important and well-documented behavioral differences between humans and neural networks. The first of these is texture-vs-shape bias which measures how much an observer relies on texture vs shape cues for object recognition. Empirically, Geirhos et al. [7] measured an observer's shape bias by examining their response during a task that involved 16-way categorization (same categories that we use) of cue-conflict images. These images contain shape information from one category and texture information from another (e.g., a cat with an elephant's texture; Fig. 3). They defined shape bias as the fraction of images in their dataset categorized by an observer according to the shape category, and measured human shape bias as 0.99, meaning they categorize almost 100% of the images by shape. For all our shape bias analyses, we used cue-conflict images from their dataset to measure the shape bias of all 76 networks. All shape bias scores were averaged across object categories to obtain a single number for each network. Additionally, we use their measurement of human shape bias (0.99) because it is reliable and replicable [36].

**Adversarial robustness (whitebox accuracy).** Adversarial robustness of a neural network measures how resistant it is to adversarial attacks, stimulus perturbations generated specifically to fool the network. To measure robustness of our networks, we generated adversarial perturbations for 1000 images (each belonging to a different category) from the ImageNet validation dataset using projected gradient descent (PGD; $\|L_{\infty}\| = 0.1$, max. iterations = 32), and measured the network's accuracy

in classifying those images. In our analysis, we use this measure, whitebox accuracy, computed using the PGD implementation from the Adversarial Robustness Toolbox [56]. Additionally, as a comparison, we also measured the accuracy of all networks on 1000-way classification of all 50,000 available unperturbed ImageNet validation set images.

## 4 Results

### 4.1 Humans recognize objects in natural images using the same spatial frequencies that they use for letters and gratings.

The classic one-octave-wide spatial frequency band has appeared frequently in human psychophysics literature, as the channel for grating detection [1], and letter recognition [2] across various stimulus conditions (alphabets, fonts, sizes, low- and high-pass noise) [3]. It is a surprising result that of all the spatial frequency information available in the stimulus, the visual system would rely on such a narrow band for recognition. However, letters and gratings are largely homogeneous objects, and it is not obvious that those results will generalize well to the recognition of diverse, heterogeneous, noisy objects in the real world. From our critical band masking analyses, we report a surprising result. Humans use the same narrow, one-octave-wide spatial frequency band to recognize objects in natural images that they use for letter and grating recognition, making this channel a canonical feature of human object recognition. In Fig. 4 we show accuracy heatmaps of humans for the various noise conditions averaged across observers (A, left), and a Gaussian channel fit to thresholds from this heatmap (B, black curve). The bandwidth (see 3.4) of the human channel is 1.00 octave, consistent with previous measurements of 1-2 octaves for grating/sinusoid detection [57], and $1.6 \pm 0.7$ octaves for letter recognition [3].

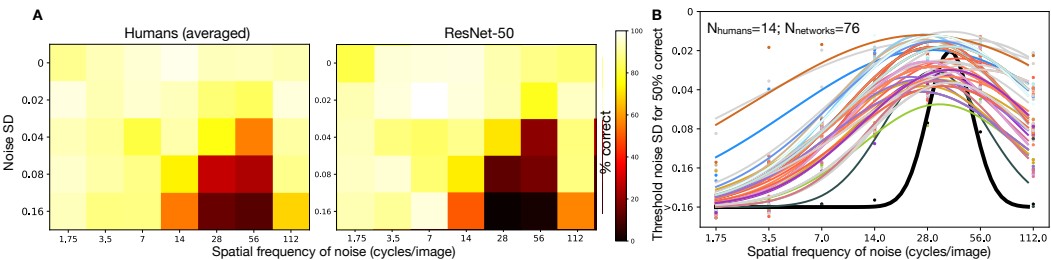

Figure 4: *Results of 14 humans and 76 neural networks on critical band masking task.* **A.** Heatmaps showing % correct of humans (averaged across 14 observers given strong cross-observer consistency, see Supplementary Material for individual heatmaps) and ResNet-50, an example network. **B.** Threshold noise SD for 50% correct (points; jittered by 5% of range for visualization) and Gaussian fit i.e. channel, of human (averaged; black curve) and all 76 tested networks (all other colors).

### 4.2 The neural network channel is 2-4 times as wide as the human channel.

Having seen the same channel in humans across a variety of stimuli and stimulus conditions, a natural next question is: how about neural-network-based object recognition systems? On repeating the same critical band masking experiment and analyses on our set of 76 neural networks (accuracy heatmap for a sample network ResNet-50 shown in Fig. 4A, right), we plot the channel for networks alongside the human one in Fig. 4B. We observe that the network channel, across the various architectures and training procedures used in our set, is ≈2-4 times as wide as the human channel (1.00 octaves), ranging from 2.30 (ViT-B) to 5.98 (resnet50-l2-eps1, an adversarially-trained network). In other words, neural networks are susceptible to noise at both high and low spatial frequencies that doesn't affect human performance.

### 4.3 Channel properties correlate strongly with shape bias, and with robustness of adversarially-trained networks.

We then ask if this stark difference between the spatial frequencies networks and humans use for recognition correlates with two other important and well-documented differences in human and

network behavior – shape bias and adversarial robustness. Humans are well-known to rely primarily on shape cues for object recognition and other visual tasks [10, 7]. Neural networks, on the other hand, show a lot of variance — convolutional networks use texture cues [7, 8], while other architectures such as transformers are shape-biased [9, 58]. Likewise, neural networks are generally strongly sensitive to tiny perturbations to their input that might be imperceptible to humans, called adversarial perturbations [6]. Fig. 5 plots each property of the spatial frequency channel against shape bias and whitebox accuracy (see 3.4 for metric details), along with line-fits (only significant fits with $p < 0.05/18$ (Bonferroni-corrected) shown).

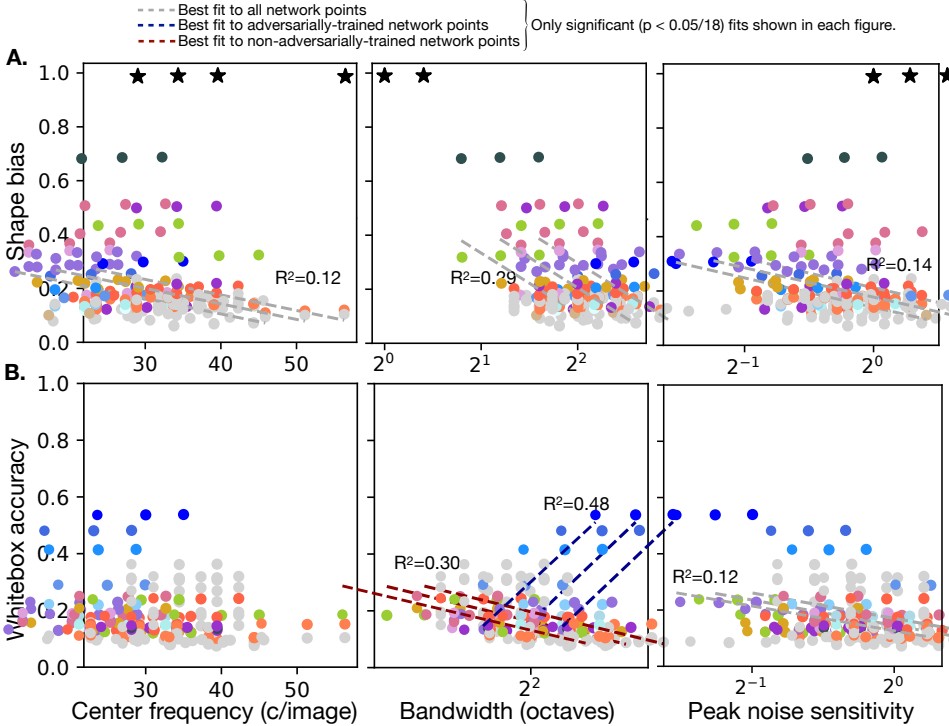

Figure 5: *Relationship between network channel properties (center frequency, bandwidth, peak noise sensitivity), shape bias and adversarial robustness.* Each figure plots one channel property against either shape bias or adversarial robustness (whitebox accuracy), for all networks represented by colored points. Adversarially-trained networks are represented by blue-colored points and increasing intensity of blue (light to dark) corresponds to increasing L2-adversarial robustness (i.e., stronger adversarial training). 3 regression lines were fit in each figure – to all network points (gray-dotted line), to only adversarially-trained networks (blue-dotted line), to all non-adversarially-trained networks (crimson-dotted line) – and only those with $p < 0.05/18$ (Bonferroni-corrected) are shown. In shape bias figures, the black star represents the human average (shape bias value taken from [36]). The three channel properties together explain 51% of variance in shape bias of all networks, and 66% of variance in whitebox accuracy across adversarially-trained networks.

Networks show a large range of shape bias scores. Although most are clustered around 0.1-0.2 (10-20% of cue-conflict images categorized by shape), a few classes of networks are more strongly shape biased: ViTs (green, violet-red), scaled-up CNNs (BiTs in purple, Noisy Student in dark gray). Channel bandwidth strongly accounts for these differences, single-handedly explaining 29% of the variance, with a slope of $-0.22 \pm 0.04$. Networks with a smaller channel bandwidth are more shape biased. This trend is also supported by the human result (black star in middle panel of Fig. 5A). Humans are the most shape biased and have the smallest bandwidth. Center frequency and peak noise sensitivity also account for some of the variance: a linear model with center frequency, bandwidth and peak noise sensitivity accounts for 51% of the cross-network variance in shape bias, which is surprising given that shape is often thought to be a complex feature of objects.

When we look at how well channel properties account for variance in whitebox accuracy (adversarial robustness), we see no such pattern (Fig. 5B). Of the three, only noise sensitivity shows a significant

fit (gray dotted line in rightmost plot in Fig. 5B). However, in these plots, we see some interesting patterns when we separately consider adversarially-trained networks (blue-colored points of any shade) and other networks. Increasing intensity of blue represents the same network architecture (ResNet-50) subjected to increasing amounts of adversarial training to yield a monotonic rise in their measured L2-adversarial robustness ($\epsilon \in \{0, 0.01, 0.03, 0.05, 0.1, 0.25, 0.5, 1, 3, 5\}$; see [51] for details). Since these points clearly differ visually in our plots from other networks, we fit two more lines, one to only adversarially-trained networks (dark blue, dotted), and the other to only the remaining (dark red, dotted). Interestingly, this yields strong fits for bandwidth and peak noise sensitivity cases. For adversarially-trained networks, the three channel properties together account for 66% of the cross-network variance in whitebox accuracy. Networks with higher bandwidth and higher peak noise-sensitivity are more adversarially robust.

As an aside, we suspect that the bimodal nature of white-box accuracy results is because unlike the case of shape bias, most networks we tested have poor adversarial robustness, and hence lie on the flat portion of its psychometric function, while strongly adversarially-trained networks lie on the steep portion. Therefore, fitting lines to these two groups separately might be akin to fitting two lines to describe a non-linear psychometric function.

### 4.4 Adversarial training expands the channel further beyond the human bandwidth

Embedded in our correlational results above is an important causal relationship. Given that the adversarially-trained networks we test differ only in terms of the amount of adversarial training (measured using L2-robustness), we can analyze the effect that the amount of adversarial training has on channel properties. Firstly, as shown in Fig. 6, increasing L2-robustness ($\epsilon$) corresponds to a monotonic increase in whitebox accuracy and a gradual decrease in ImageNet validation set accuracy. Secondly, we observe that as L2-robustness and whitebox accuracy of adversarially-trained networks increase, their bandwidth increases too, moving away from the human bandwidth. This means that networks that are more robust to adversarial attacks (according to both L2-robustness and whitebox accuracy) are in fact more sensitive to noise added in a larger set of spatial frequency bands that humans are not sensitive to. Thus, adversarial training in effect, makes them more susceptible to another kind of adversarial attack (filtered noise perturbation).

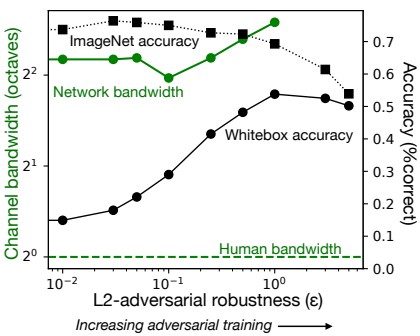

Figure 6: *Effect of adversarial training on network channel bandwidth.* ImageNet validation set accuracy (black square-dotted line), whitebox accuracy (black circle-solid line) of adversarially-trained networks with increasing L2-robustness (increasing amount of adversarial training), network channel bandwidth (green circle-solid line), and human bandwidth value (green dotted line). As robustness increases, whitebox accuracy improves at the cost of both ImageNet accuracy and expansion of the network bandwidth further beyond the human bandwidth.

## 5 Discussion

Having shown that neural network spatial-frequency channels diverge from the canonical one-octave-wide channel found in humans, we propose that critical band masking should be added to the toolbox of metrics for model-human comparison. Our dataset of 14 humans and 76 networks on 16-way categorization raises important questions about which spatial frequencies matter for object recognition.

Every one of the 76 networks we tested has a wide critical band. We tested all the networks from Geirhos et al. [36] which is the largest comparison of networks and humans on object recognition. That study included most of the popular network kinds, spanning a wide range of conditions: convolutional networks and transformers, shallow and deep networks, supervised and self-supervised training, standard and adversarially-trained networks. Thus, the conclusions of this paper are based on a representative sample of popular network designs. More work is required to explore the effects of diverse kinds of training data and augmentation.

When comparing biological vision and artificial neural networks, there are two questions to be asked: what can artificial networks learn from biology, and how can neural networks inform our understanding of biology [59, 34]. Needless to say, the origins of deep learning relied heavily on knowledge of biology [30]. But can biology impact the evolution of neural networks? Dapello et al. [35] showed that making the first layer's tuning match biological V1 tuning improves robustness. The critical band idea is central to understanding hearing and vision. Here we show that the critical bands, which are a measure of tuning, are different between humans and networks, and that this difference strongly explains robustness (shape bias and adversarial robustness) of networks. Thus, the results presented here are evidence that narrowing the machine critical band to match that of humans may make neural networks even more robust.

The critical band masking paradigm characterizes the object recognition channel by measuring sensitivity to frequency-filtered noise [4, 2]. Our experiments use this approach to study how spatial frequency selectivity relates to noise robustness, a core topic in machine learning research. Indeed, Geirhos et al. [36] found that adversarial training of networks in the presence of high-frequency noise decreases their overall noise sensitivity. An alternative approach determines what spatial frequencies are used to recognize objects, by filtering the image itself, i.e., removing some frequencies instead of adding filtered noise to mask them [60, 18, 36]. The two approaches yield different results in humans [3], which we will explore in future work.

Li et al. [18] found that adversarially-robust object recognition models rely mainly on low spatial-frequency information in images (also see [15]). Our observations qualitatively agree with their results. Our network channel is indeed centered on a low-frequency band, and adversarial training expands its bandwidth. More work is needed to quantitatively compare the two approaches.

# 6 Conclusions

In this paper, we introduce the critical band masking paradigm to study what spatial frequency information humans and neural networks use to recognize objects in natural images. To this end, we use a noise-masked ImageNet object recognition task. First, we find that humans use the same canonical one-octave-wide channel to recognize objects in natural images that they use for letter and grating recognition. Second, the neural network channel is 2-4 times wider than the human channel i.e., network performance is impaired by high and low spatial frequency noise to which humans are immune. Furthermore, properties of the spatial frequency channel correlate strongly with shape bias (explaining 51% cross-network variance) and robustness of adversarially-trained networks (explaining 66% variance). Finally, we find that adversarial training increases robustness but further widens the already-too-wide network channel. Overall, critical band masking allows systematic frequency-based analysis of both network and human object recognition performance, revealing important differences in noise sensitivity. The critical band offers a spatial-frequency-based explanation of shape bias and adversarial robustness. Thereby, our paper provides evidence suggesting that efforts to make a network more robust should look for ways to narrow its critical band. While our results are based on a large, representative sample of popular network designs, more work is required to explore the effects of diverse kinds of training data and augmentations. Such experiments would also help determine if the relationship between channel properties and robustness is also causal.

## Acknowledgments and Disclosure of Funding

We thank Furkan Özçelik and Ozhan Dehghani for ongoing work, and Eero Simoncelli, James Elder, Maria Pombo, Paul Gavrikov, our five anonymous reviewers, and audience at VSS, COSYNE, and ECVP for helpful comments and suggestions. A.S. additionally thanks Jiaming Xu and Chianna Cohen for being dinner companions during the pre-deadline slog. Our work was funded by grants from the National Eye Institute, National Institutes of Health (R01-EY027964 to D.G.P. and R01-EY031446 to N.J.M.). It was also supported by a National Institutes of Health core vision grant (P30-EY013079). We are also grateful to NYU's High Performance Computing (HPC) cluster for enabling our neural network experiments and analysis.

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
