# Spatial frequency channels, shape bias, and adversarial robustness
# —Supplementary material—

## A   Human psychophysics

Fig. 1 shows screenshots from our online psychophysical critical band masking experiment. Accuracy heatmaps computed for different observers in our experiment showed little individual difference (Fig. 2) and an even smaller difference in terms of threshold noise SD for 50% accuracy. Table 1 shows the value of each channel property computed from Gaussian fits to the averaged human data versus those found by summarizing Gaussian fits to individual human data. Given that they are similar for all channel properties, we use the former for all reported human data in the main paper.

Our existing method for computing thresholds and fitting the Gaussian function to them is difficult to apply to observers that have very high noise sensitivity (low efficiency) since it relies on good performance for the baseline (zero-noise) condition. In section F, we describe an alternative approach that would enable us to derive channels for humans and/or networks with low overall noise sensitivity.

| Channel property | from "Fit to average" | from "Average of fits" |
|---|---|---|
| Bandwidth | 1.00 | $1.43 \pm 0.54$ |
| Center frequency | 39.60 | $39.13 \pm 2.98$ |
| Peak noise sensitivity | 1.00 | $0.80 \pm 0.28$ |

Table 1: *Comparison between channel properties computed from Gaussian fit to the averaged human data ("Fit to average"), and Gaussian fit to individual human data ("Average of fits").* We use the former for all reported human data in the main paper.

## B   Neural network evaluation

We tested 76 neural networks on the same critical band masking task that we used for humans. These networks spanned a large range of architectures and pretraining procedures. In our paper, we divide them into classes and color code each. Figure 3 illustrates the color used for each network we tested. The classes of networks we considered spanned standard convolutional networks (ResNets, ResNeXts, VGGs, Densenets, etc.), scaled-up convolutional networks (BiT-M, SWSL, Noisy Student), vision transformers (ViTs), self-supervised networks (SimCLRs), adversarially-trained networks (resnet50_l2_eps$\epsilon$, where $\epsilon \in \{0, 0.01, 0.03, 0.05, 0.1, 0.25, 0.5, 1, 3, 5\}$), networks with *human-inspired* training (ResNet-50 trained on stylized-ImageNet, VOneNet), and miscellaneous other networks (DPNs, bagnets, HRNets, etc.).

## C   Fitting Gaussian function to thresholds

To obtain network and human channels, we fit a Gaussian function to threshold noise strengths for 50% accuracy at each spatial frequency band. The Gaussian function we fit had three parameters: peak height ($A$), mean ($\mu$), and standard deviation ($\sigma$).

37th Conference on Neural Information Processing Systems (NeurIPS 2023).

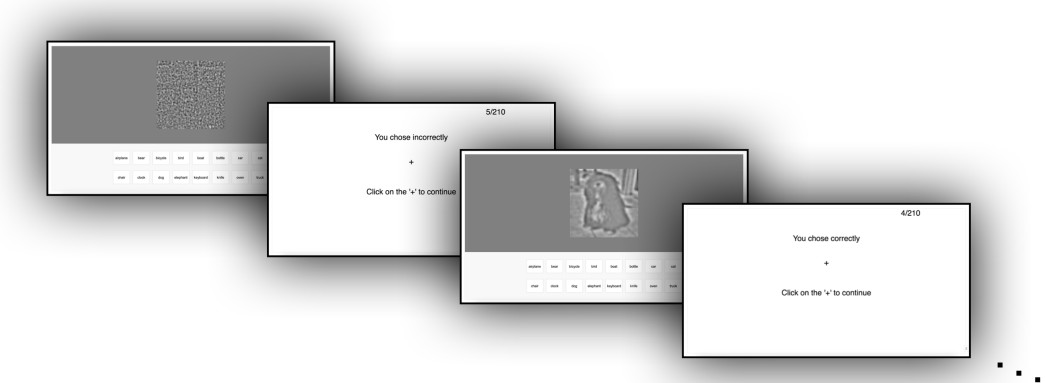

Figure 1: *Screenshots from our human psychophysics experiment.* On each trial, the observer was presented with a noise-masked ImageNet image (see Section 2.1 of main paper for image generation details.) on a uniformly gray background and 16 buttons below it representing the various object categories. On selecting the button that they thought best described the object in the image, they would be presented with feedback (correct or incorrect), and a fixation cross that they were instructed to click to move to the next trial.

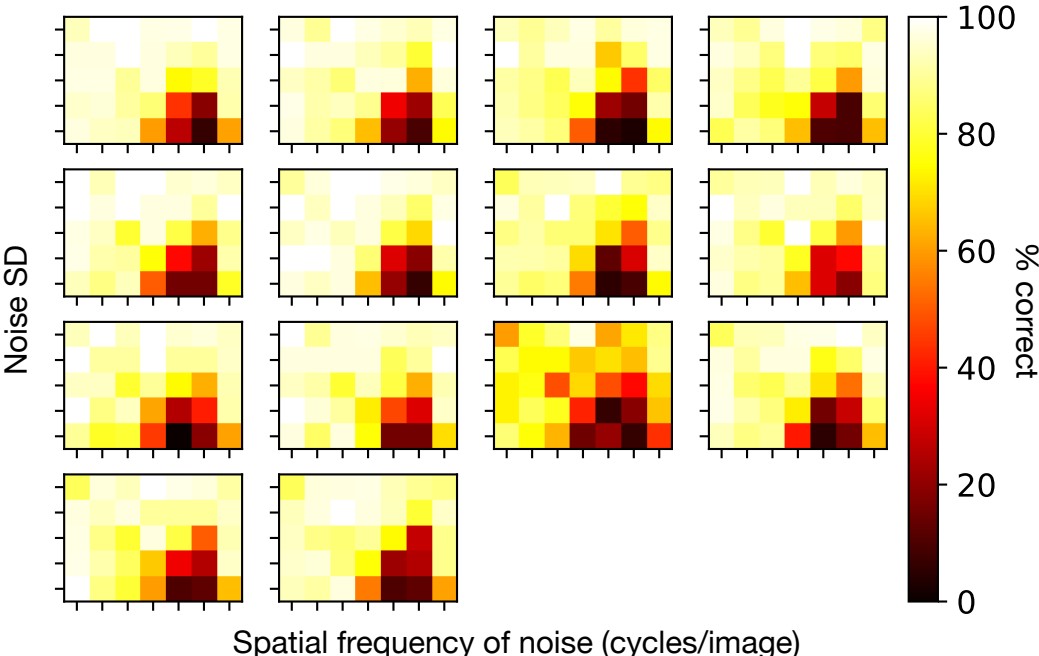

Figure 2: *Accuracy heatmaps for each observer in our human psychophysics experiment.* People showed remarkably low variance in their accuracy for each condition, and an even smaller variance in their 50% accuracy thresholds computed for each spatial frequency condition.

$$f(x) = Ae^{-(x-\mu)^2/(2\sigma^2)}$$

Across humans and all 76 networks, we were able to obtain good fits (with low standard error) for all 3 parameters. Table 2 shows the best-fit value of each parameter along with corresponding standard errors. Once these parameters are fit to each set of thresholds, we compute the three channel properties (center frequency, bandwidth, and peak noise sensitivity) using them.

$$\text{Bandwidth (octaves)} = 2\sigma\sqrt{\ln 4}$$
$$\text{Center frequency (cycles/image)} = 1.75 \times 2^{\mu}$$
$$\text{Peak noise sensitivity (per unit SD)} = 2^{A-4}$$

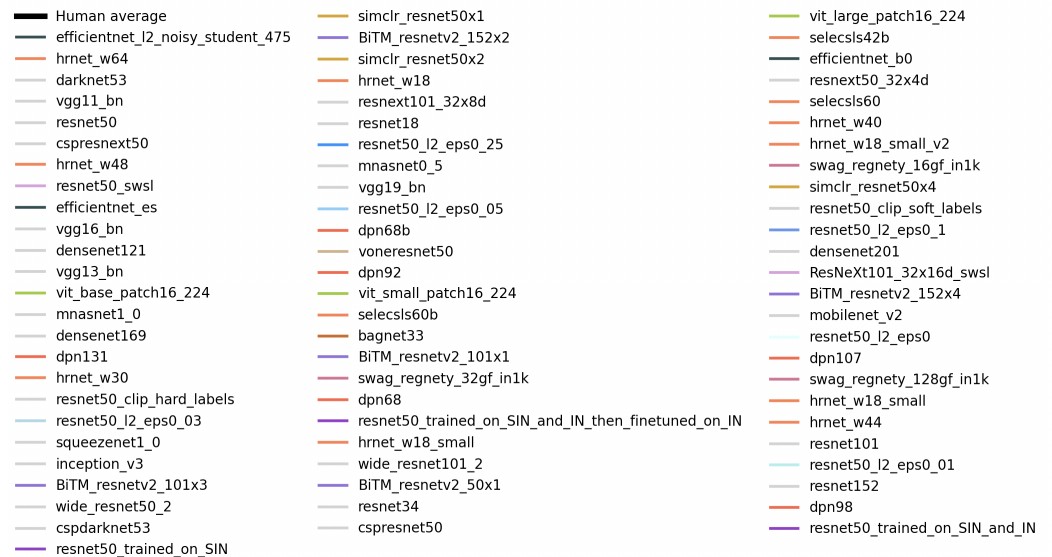

Figure 3: *Color code used for humans (averaged) and each network tested in our experiment.* We use averaged data for humans because they showed remarkably low variance in their accuracy for each condition, and an even smaller variance in their 50% accuracy thresholds computed for each spatial frequency condition.

## D    Preventing edge-case image clipping

Even after contrast-reduction of the grayscale ImageNet images to ≈20% of original, there were 2 images in our dataset for which 20 pixels of the $224 \times 224$ were still clipped for the high noise (0.16), high-frequency (112.0 cycles/image) condition. Although this is a very small fraction of clipped pixels, we wanted to verify that the spatial frequency information still lay within the required octave-wide band. To prevent this clipping, we generated the noise, and then distorted the image to prevent pixel clipping and then added back the same noise. This ensured that the distribution of noise still remained Gaussian. However, this is no different from clipping. So we additionally verified, using basic fourier analysis, that the frequencies present in the noise were indeed in the 112.0-224.0 cycles/image frequency band.

## E    How do grayscale-conversion and contrast-reduction affect network performance?

The pretrained networks we tested were predominantly trained on colored ImageNet images. For the sake of critical band masking, we converted ImageNet images to grayscale and then reduced them to 20% contrast. This represents a testing condition outside of the training data domain of the networks. Therefore, it is important to verify that our transformation does not severely affect test-time performance.

Fig. 4 shows the percent correct values of each network we tested for each of 3 image conditions: color, gray, and low-contrast gray. Of the 88 networks available in our source datasets, 12 were below threshold (50% accuracy) for low-contrast gray images (alexnet, squeezenet1_0, squeezenet1_1, shufflenet_v2_x0_5, bagnets 9, 17, 33, resnet50_l2_eps1,3,5, selecsls42b, selectls84), which made

| Observer | $A$ (mean) | $A$ (SE) | $\mu$ (mean) | $\mu$ (SE) | $\sigma$ (mean) | $\sigma$ (SE) |
|---|---|---|---|---|---|---|
| human | 4.00 | 0.25 | 4.50 | 0.00 | 0.42 | 0.02 |
| efficientnet_l2_noisy_student_475 | 3.49 | 0.48 | 4.20 | 0.15 | 0.97 | 0.15 |
| hrnet_w64 | 3.72 | 0.45 | 4.29 | 0.26 | 1.70 | 0.30 |
| darknet53 | 3.51 | 0.48 | 4.49 | 0.31 | 1.69 | 0.36 |
| vgg11_bn | 4.27 | 0.17 | 4.95 | 0.22 | 2.29 | 0.21 |
| resnet50 | 3.51 | 0.48 | 4.49 | 0.31 | 1.69 | 0.36 |
| cspresnext50 | 3.18 | 0.11 | 4.49 | 0.07 | 1.55 | 0.08 |
| hrnet_w48 | 4.24 | 0.31 | 4.21 | 0.13 | 1.54 | 0.15 |
| resnet50_swsl | 3.51 | 0.48 | 4.49 | 0.31 | 1.69 | 0.36 |
| efficientnet_es | 4.24 | 0.31 | 4.21 | 0.13 | 1.54 | 0.15 |
| vgg16_bn | 4.38 | 0.25 | 4.25 | 0.15 | 1.89 | 0.17 |
| densenet121 | 4.11 | 0.13 | 4.46 | 0.10 | 1.96 | 0.11 |
| vgg13_bn | 4.27 | 0.17 | 4.95 | 0.22 | 2.29 | 0.21 |
| vit_base_patch16_224 | 2.63 | 0.39 | 4.30 | 0.27 | 1.49 | 0.30 |
| mnasnet1_0 | 4.38 | 0.25 | 4.25 | 0.15 | 1.89 | 0.17 |
| densenet169 | 4.09 | 0.24 | 4.15 | 0.12 | 1.71 | 0.14 |
| dpn131 | 3.59 | 0.36 | 4.22 | 0.26 | 1.91 | 0.30 |
| hrnet_w30 | 3.59 | 0.36 | 4.22 | 0.26 | 1.91 | 0.30 |
| resnet50_clip_hard_labels | 4.02 | 0.44 | 4.11 | 0.50 | 2.72 | 0.63 |
| resnet50_l2_eps0_03 | 4.41 | 0.26 | 4.00 | 0.12 | 1.66 | 0.13 |
| squeezenet1_0 | 4.25 | 0.18 | 4.21 | 0.38 | 3.76 | 0.55 |
| inception_v3 | 4.41 | 0.26 | 4.00 | 0.12 | 1.66 | 0.13 |
| BiTM_resnetv2_101x3 | 2.95 | 0.38 | 3.92 | 0.28 | 1.77 | 0.31 |
| wide_resnet50_2 | 3.69 | 0.30 | 4.59 | 0.14 | 1.42 | 0.16 |
| cspdarknet53 | 3.69 | 0.30 | 4.59 | 0.14 | 1.42 | 0.16 |
| resnet50_trained_on_SIN | 3.39 | 0.31 | 4.27 | 0.17 | 1.57 | 0.19 |
| simclr_resnet50x1 | 3.27 | 0.42 | 4.16 | 0.38 | 2.09 | 0.44 |
| BiTM_resnetv2_152x2 | 2.95 | 0.38 | 3.92 | 0.28 | 1.77 | 0.31 |
| simclr_resnet50x2 | 2.95 | 0.38 | 3.92 | 0.28 | 1.77 | 0.31 |
| hrnet_w18 | 4.11 | 0.13 | 4.46 | 0.10 | 1.96 | 0.11 |
| resnext101_32x8d | 3.18 | 0.11 | 4.49 | 0.07 | 1.55 | 0.08 |
| resnet18 | 4.11 | 0.13 | 4.46 | 0.10 | 1.96 | 0.11 |
| resnet50_l2_eps0_25 | 4.02 | 0.44 | 4.11 | 0.50 | 2.72 | 0.63 |
| mnasnet0_5 | 4.48 | 0.39 | 4.51 | 0.35 | 2.27 | 0.38 |
| vgg19_bn | 4.09 | 0.24 | 4.15 | 0.12 | 1.71 | 0.14 |
| resnet50_l2_eps0_05 | 4.41 | 0.26 | 4.00 | 0.12 | 1.66 | 0.13 |
| dpn68b | 4.24 | 0.31 | 4.21 | 0.13 | 1.54 | 0.15 |
| voneresnet50 | 4.29 | 0.34 | 3.97 | 0.17 | 1.79 | 0.20 |
| dpn92 | 3.80 | 0.39 | 4.66 | 0.34 | 1.96 | 0.36 |
| vit_small_patch16_224 | 3.09 | 0.34 | 4.36 | 0.27 | 1.79 | 0.31 |
| selecsls60b | 4.09 | 0.24 | 4.15 | 0.12 | 1.71 | 0.14 |
| bagnet33 | 4.40 | 0.51 | 3.98 | 0.70 | 3.28 | 1.02 |
| BiTM_resnetv2_101x1 | 3.33 | 0.27 | 3.75 | 0.16 | 1.67 | 0.17 |
| swag_regnety_32gf_in1k | 3.59 | 0.53 | 3.95 | 0.30 | 1.68 | 0.33 |
| dpn68 | 4.09 | 0.24 | 4.15 | 0.12 | 1.71 | 0.14 |
| resnet50_trained_on_SIN_and_IN_then_finetuned_on_IN | 3.59 | 0.36 | 4.22 | 0.26 | 1.91 | 0.30 |
| hrnet_w18_small | 4.11 | 0.13 | 4.46 | 0.10 | 1.96 | 0.11 |
| wide_resnet101_2 | 3.18 | 0.11 | 4.49 | 0.07 | 1.55 | 0.08 |
| BiTM_resnetv2_50x1 | 3.28 | 0.18 | 4.04 | 0.14 | 1.94 | 0.16 |
| resnet34 | 3.75 | 0.28 | 4.34 | 0.22 | 2.03 | 0.25 |
| cspresnet50 | 3.69 | 0.30 | 4.59 | 0.14 | 1.42 | 0.16 |
| vit_large_patch16_224 | 3.22 | 0.22 | 4.22 | 0.10 | 1.29 | 0.11 |
| selecsls42b | 3.52 | 0.30 | 5.01 | 0.49 | 2.34 | 0.47 |
| efficientnet_b0 | 4.09 | 0.24 | 4.15 | 0.12 | 1.71 | 0.14 |
| resnext50_32x4d | 3.65 | 0.37 | 4.92 | 0.40 | 1.97 | 0.41 |
| selecsls60 | 3.59 | 0.36 | 4.22 | 0.26 | 1.91 | 0.30 |
| hrnet_w40 | 3.34 | 0.43 | 4.45 | 0.39 | 1.98 | 0.43 |
| hrnet_w18_small_v2 | 4.09 | 0.24 | 4.15 | 0.12 | 1.71 | 0.14 |
| swag_regnety_16gf_in1k | 3.72 | 0.58 | 4.04 | 0.28 | 1.51 | 0.30 |
| simclr_resnet50x4 | 3.10 | 0.43 | 4.04 | 0.25 | 1.53 | 0.28 |
| resnet50_clip_soft_labels | 4.11 | 0.13 | 4.46 | 0.10 | 1.96 | 0.11 |
| resnet50_l2_eps0_1 | 4.25 | 0.28 | 4.26 | 0.20 | 2.07 | 0.23 |
| densenet201 | 3.59 | 0.36 | 4.22 | 0.26 | 1.91 | 0.30 |
| ResNeXt101_32x16d_swsl | 3.72 | 0.58 | 4.04 | 0.28 | 1.51 | 0.30 |
| BiTM_resnetv2_152x4 | 2.95 | 0.38 | 3.92 | 0.28 | 1.77 | 0.31 |
| mobilenet_v2 | 3.52 | 0.30 | 5.01 | 0.49 | 2.34 | 0.47 |
| resnet50_l2_eps0 | 4.38 | 0.25 | 4.25 | 0.15 | 1.89 | 0.17 |
| dpn107 | 3.09 | 0.34 | 4.36 | 0.27 | 1.79 | 0.31 |
| swag_regnety_128gf_in1k | 3.22 | 0.22 | 4.22 | 0.10 | 1.29 | 0.11 |
| hrnet_w18_small | 4.11 | 0.13 | 4.46 | 0.10 | 1.96 | 0.11 |
| hrnet_w44 | 3.59 | 0.36 | 4.22 | 0.26 | 1.91 | 0.30 |
| resnet101 | 3.51 | 0.48 | 4.49 | 0.31 | 1.69 | 0.36 |
| resnet50_l2_eps0_01 | 4.11 | 0.13 | 4.46 | 0.10 | 1.96 | 0.11 |
| resnet152 | 3.18 | 0.11 | 4.49 | 0.07 | 1.55 | 0.08 |
| dpn98 | 3.51 | 0.48 | 4.49 | 0.31 | 1.69 | 0.36 |
| resnet50_trained_on_SIN_and_IN | 3.51 | 0.48 | 4.49 | 0.31 | 1.69 | 0.36 |

Table 2: Best Gaussian channel fit parameters (mean) and standard errors (SE) for humans and all 76 networks.

them impossible to test on our critical band masking task. This left us with the 76 networks that we report results for in the main paper. Importantly, for these networks, we see in the figure that the change in performance from the color to gray to low-contrast gray condition is very small.

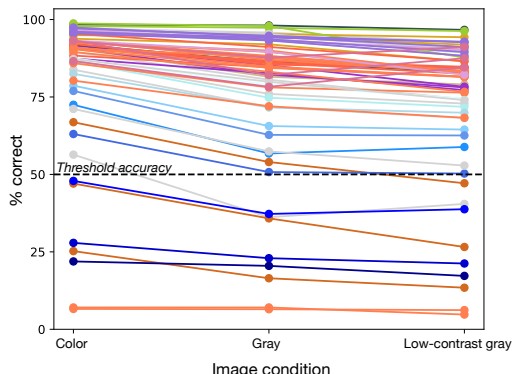

Figure 4: *Effect of grayscale-conversion and contrast-reduction on network performance.* Each colored line represents one of the 88 networks in our source datasets. For each network, we plot three points, one for each image condition. 'Color' refers to standard ImageNet, 'Gray' to grayscale-converted ImageNet, and 'Low-contrast gray' to grayscale-converted and contrast-reduced ImageNet. The horizontal dotted-line at 50% is the threshold accuracy we use when computing the spatial-frequency channel. Therefore, a basic requirement for our analysis is that networks perform above-threshold for the zero-noise condition ('low-contrast gray' here)

## F   Normalized thresholds: an alternative approach

Here, we present an alternative approach (that we did not use in the paper) to calculating thresholds that determine the spatial-frequency channel. In the approach we use in the paper, we calculate the noise SD required for accuracy in each spatial frequency condition to drop below 50%. This relies on the fact that the accuracy of a model on the baseline condition (low-contrast grayscale images) is above 50%. Of the 86 networks available in our source set of networks, we were able to test 76 because the remaining did not satisfy above threshold accuracy on the baseline condition. This is sensible because any reasonably good model of human vision should achieve more than 50% accuracy on low-contrast gray images that humans score well above 80% on. But if one wanted to nevertheless calculate network thresholds, we propose normalizing accuracy of their heatmap with respect to their accuracy in the baseline condition. Then, we can calculate the threshold noise SD for 50% normalized accuracy, and fit Gaussian functions to that. Testing this on a fraction of our networks, we find that this modified method yields channels that are very similar to those obtained using the unnormalized method we use in the paper, with the added benefit that we could also compute inverted-U-shaped Gaussian fits for the networks that failed our original analysis. We do not include this method in the main paper because normalized accuracy is not comparable across networks because each network's accuracy will be normalized independently.

## G   Robustness evaluation with commonly used parameters

To evaluate network robustness in the main paper, we used a 32-step PGD attack with $\|L_\infty\| = 0.1$. Here, we present the same analysis with more commonly used parameter values, 20-step PGD with $\|L_\infty\| = 4/255$.

For non–adversarially trained networks, we see the same trend as before – bandwidth is lower for networks with higher whitebox accuracy. But for adversarially trained networks, we no longer see what we saw with our previous attack parameters – there is no significant correlation (with bonferroni correction) between whitebox accuracy and channel bandwidth.

We think this absence of correlation is mainly because the new attack is much weaker than our old attack, and so we see almost no difference in whitebox accuracy for networks with drastically different amounts of adversarial training. In other words, even relatively weak adversarial training is able to make networks robust to the new attack.

## H   Compute details

To evaluate all networks, we ran CUDA-enabled PyTorch implementations on either a NVIDIA V100 or an RTX8000 GPU. Testing all 76 networks on our critical band masking task took roughly 4-5 hours. Shape bias and adversarial robustness calculation took another 5 hours and 6 hours respectively.