# OpenReview forum: "Spatial-frequency channels, shape bias, and adversarial robustness"
_NeurIPS.cc/2023/Conference — NeurIPS 2023 oral_

### Official Review · Reviewer_TCTa · 2023-06-14

**Soundness:** 2 fair
**Presentation:** 4 excellent
**Contribution:** 4 excellent
**Rating:** 7
**Confidence:** 5

**Summary:**

This study applies a well-established procedure from psychophysics, critical band masking, to an object recognition task on which human and a wide range of neural networks are compared.
The core finding is that while humans use roughly the same spatial frequency channels for object recognition as they do for e.g. letter / grating recognition, all 76 investigated neural networks use much wider channels (more than twice as wide as the human channel).
This means that humans and neural networks rely on different spatial frequencies for object recognition.

**Strengths:**

* Overall, I really liked this study - it has a simple yet effective protocol, clear findings, and approaches a novel question through sound experiments.
* The proposed approach is applicable to the analysis of future networks, i.e. it is sufficiently general such that it can be used by follow-up studies / on general architectures
* Successful transfer of psychophysical method to the study of neural networks
* The paper is very well-written, easy to follow, clearly structured and has good visualizations
* Human data will be made publicly available upon publication

**Weaknesses:**

Significance testing:
* Figure 6 does not correct for multiple comparisons. I would like to know which of the plotted effects still hold after Bonferroni correction by dividing the alpha level (.05) by 18, i.e. the number of comparisons (6 plots x 3 fits), resulting in a corrected alpha level of 0.0027777. I would encourage the authors to update the figure with a corrected version.

Human experimental weaknesses:
* In human visual perception, spatial frequency is measured in cycles per degree of visual angle and depends on the distance to the monitor. If I understand correctly, the setup is employed with unknown distance between observers and monitor.
* No information on monitor calibration, which is very important in experiments where stimuli of reduced contrast are shown. E.g., many monitors have settings that "enhance" contrast, which is not unlikely to have an impact on human experimental performance. Did the authors take this into account?
* Human experiments with reduced contrast are best performed in a setting where contrast adaptation can take place. In the present study, however, the screen constantly changes from grey to bright white and back to grey again.
* Given the online study setup, how did the authors ensure normal / corrected-to-normal vision (beyond self-reporting, where online participants have monetary incentives for inaccurate reporting)?
* Two of the 16 observers (supplementary Figure 2) do not seem to perform well at all on the simple task

**Questions:**

### Questions:
* What is going on with the two outlier observers in Figure 2 (1st column, bottom two rows)?
* Given that some networks have severely reduced performance at 20% contrast, I'm wondering whether there might be a possible interaction of the experimental findings with the contrast level. As a supplementary experiment, it might be interesting to test the core findings separately for (a) networks which only show a minor performance detriment when reducing images to 20% contrast and (b) the rest of the networks.

### Major opportunities for improvement:
(The study is already quite solid, and I am in favor of publication. These suggestions go beyond what is currently done in the paper. If at least one of them can be addressed, I would be happy to increase my score further.)
* "we propose that critical band masking should be added to the toolbox of metrics for model-human comparison.": I agree, and I was wondering whether the authors plan to facilitate this. For instance, they could release their stimuli as part of an open-source testbed (e.g., as a fork of  / addition to the model-vs-human toolbox, or in any other way).
* The authors show that properties of critical bands are correlated with important properties like shape bias / adversarial robustness. Going beyond a correlation, I was wondering whether a causal link could be established, e.g. by reducing the signal-to-noise ratio of non-human channels in the input during training, leading to changed critical bands (e.g., more human-like ones), and then showing that the properties of interest (shape bias / adversarial robustness / ...) change accordingly.
* The reasons for the human-model divergence are not yet understood. It would be interesting to know why this divergence arises, and whether it is a property of the architecture, the training procedure, or something else.


### Minor questions:
* line 86: should this be "human object recognition"?
* why "roughly equally distributed across all 16 categories"? What was the exact distribution, and why was it not exactly equally distributed? (Background: performance can be above chance level 1/16 when categories aren't exactly equally distributed)


**Limitations:**

The authors don't have a section on limitations. In accordance with the checklist, the authors are encouraged to describe the limitations of their approach, in particular related to the experimental choices mentioned in the "weaknesses" section above.

---

> ### Author Rebuttal · Authors · 2023-08-10
>
> **Strengths:**
>
> >Overall, I really liked this study...
> Thank you!
>
> **Weaknesses:**
>
> >Significance testing:
>
> >Figure 6 does not correct for multiple comparisons...
>
> Done. We applied the correction to our results and do not see any change in any of the results discussed in the paper. There were a few minor changes however, which are reflected in the modified figure Fig. 3 shown in the attached PDF.
>
> >Human experimental weaknesses:
>
> >In human visual perception, spatial frequency is...
>
> Good point. We estimate that the viewing distance of online observers varies over a 2:1 range, 40 to 80 cm because <40 requires an uncomfortable accommodative effort and >80 makes it impossible to reach a laptop keyboard. However, human vision, to a first approximation, scales with image size. Majaj et al. 2002 measured the tuning of spatial frequency channels as a function of letter size and found a slight deviation whereby the channel frequency slightly less-than-doubles ($2^{\frac{2}{3}}$) when the letter size is halved. Thus, a large 2:1 variation in viewing distance will produce a small ($2^{\frac{1}{3}}=1.26:1$) variation in channel center frequency in cycles per image. Thus, the uncontrolled variation in viewing distance contributed only slightly to the 2:1 bandwidth that we measured.
>
> >No information on monitor calibration...
>
> We plan to add the following to Methods.
>
> *Nearly all of our online observers will have been looking at digital displays. The ICC specified standards for color management that require a provision of an ICC profile characterizing the properties of each display (including gamma and color primaries). The HTML <img command in all modern browsers invokes color management (if enabled) to display images faithfully. Thus, the computer screens will have varied in luminance from backgrounds of 200 to 500 nits but we expect contrast to have been accurate.*
>
> *Furthermore, even if the display gamma is other than assumed (because color management was not enabled), the too high or too low contrast will affect both signal and noise equally with practically no effect on signal-to-noise ratio (SNR). It is well established that the human energy threshold grows linearly with noise power spectral density so that when the noise is strong, doubling or halving display contrast hardly affects target recognition (Pelli & Farrell, 1999).*
>
> >Human experiments with reduced contrast...
>
> Yes, it is true that the change of background luminance will change the level of light adaptation. However, our results depend on contrast, not luminance, and so will not be affected.
>
> >Given the online study setup, how did the authors...
> We did ask our participants if they have normal or corrected-to-normal vision. In principle, one could measure acuity as a check. However, this is how most online vision studies are done.
>
> >Two of the 16 observers (supplementary Figure 2)...
>
> Yes, good point. We have removed those two observers (9 and 13) from all analyses in the paper. Our main results remain the same.
>
> **Questions:**
>
> >What is going on with the two outlier...
>
> Thanks for pointing this out. Answered above.
>
> >Given that some networks have severely...
>
> Fig. 2 from the attached PDF will be added to the paper’s supplementary material and will be referenced in the main paper. This figure shows the percent correct values of each network we tested for each of 3 image conditions: color, gray, and low-contrast gray. Of the 88 networks available in our source datasets, 12 were below threshold for low-contrast gray images, which made them impossible to test on our critical band masking task. For the remaining 76, the change in performance from the color to gray to low-contrast gray condition is very small.
>
> >Major opportunities for improvement:
>
> >(The study is already quite solid...
>
> >"we propose that critical band masking...
>
> Yes, we strongly intend to do so in the near future.
>
> >The authors show that properties of...
>
> >The reasons for the human-model divergence...
>
> Indeed, we are very interested in examining how our results are influenced by training settings, and this is something we intend to pursue next. We will modify our Discussion and Conclusion sections as follows.
>
> *DISCUSSION: Every one of the 76 networks we tested has a wide critical band. We tested all the networks from Geirhos et al., 2021 which is the largest comparison of networks and humans on object recognition. The Geirhos study included most of the popular network kinds, spanning a wide range of conditions: convolutional networks and transformers, shallow and deep networks, supervised and self-supervised training, standard and adversarially-trained networks. Thus, the conclusions of this paper are based on a representative sample of popular network designs. More work is required to explore the effects of diverse kinds of training data and augmentation.*
>
> *CONCLUSION: … are based on a representative sample of popular network designs. More work is required to explore the effects of diverse kinds of training data and augmentation.*
>
> >Minor questions:
>
> >line 86...
>
> This will be corrected in the revised manuscript, thanks.
>
> >why "roughly equally distributed...
>
> We say roughly distributed because the number of images in our testbed (1100) is not exactly divisible by 16. So we distribute 1088 (16 * 68) images equally and the remaining 12 are chosen at random from all categories. We will modify Methods to include this detail.
>
> **Limitations:**
>
> >The authors don't have a section on limitations...
>
> Done. We have added a discussion of limitations to the Conclusion section. Copied below.
>
> *While our results are based on a large, representative sample of popular network designs, more work is required to explore the effects of diverse kinds of training data and augmentations. Additionally, our current results are purely correlational, and such experiments would also help determine if the relationship between channel properties and robustness is also causal.*

---

> > ### Comment · Reviewer_TCTa · 2023-08-11
> > **Thanks**
> >
> > I would like to thank the authors for their detailed reply.
> >
> > I particularly appreciated that they corrected the alpha level for multiple comparisons and provided the updated plot, and explanations regarding experimental methods. Overall, I am satisfied with the author's response and promised edits to the paper. Some experimental weaknesses remain - e.g. in terms of monitor calibration hoping is great, measuring is better - but those are inherent to the online study design and cannot be expected to be resolved in a rebuttal. My final score is "7: Accept: Technically solid paper, with high impact on at least one sub-area, or moderate-to-high impact on more than one areas, with good-to-excellent evaluation, resources, reproducibility, and no unaddressed ethical considerations." and I strongly support publication at NeurIPS.

---

### Official Review · Reviewer_vBv1 · 2023-07-04

**Soundness:** 3 good
**Presentation:** 3 good
**Contribution:** 4 excellent
**Rating:** 7
**Confidence:** 5

**Summary:**

The authors propose to measure the differences in how humans and models rely on spectral information to recognize objects by using critical band masking to ablate. They conduct a human study revealing that humans detect objects with the same channels they use to detect letters and grating. On the other hand, models show a significantly wider channel. They further correlate the properties of the channel to robustness and shape bias, e.g. showing that AT results in an even wider channel.

**Strengths:**

### Originality
The paper explores a less common yet highly important research direction. Some "model" findings were partially reported in literature, but for the most part the findings are highly original.

### Quality
The submission is of high quality (minor weaknesses see below). The authors rigorously conduct (well-documented) human experiments and benchmark 76 ImageNet models to arrive at their conclusions.

### Clarity
The paper is very well written. Key points are communicated clearly through writing and figures.

### Significance
The authors share many findings, that may be of great interest to the NeurIPS community. In particular, they measure the human critical channel and show that models measure higher bandwidths. Adversarial Training increases the channel even further. Generally, they find correlations between channel properties and shape-bias/robustness that may lead to a new perspective on robustness but also shows another misalignment between humans and models.


**Weaknesses:**

### Method
- Evaluation only on ImageNet: I never thought I would make this as a weakness but here we go ... previous work has shown that the frequency band of adversarial attacks is not always HF (the authors arrive at the same conclusion in L230ff) but also highly depends on the dataset [2,3,4]. Given that this paper "only" presents an empirical evaluation I am not sure if the findings (about models and in particular robustness) scale to other datasets. At the very least this is a limitation that should be discussed.
- Poor choice of robust networks: The authors choose $\ell_2$-AT trained ResNets as robust networks and attack them with $\ell_\infty$ attacks. However, AT is notoriously poor at generalizing to new attacks. A better choice would have been $\ell_\infty$-AT trained ResNets. There may also be some differences in the evaluation as [1] showed that the norm differently affects performance on high/lowpass data. I don't expect this to flip observations, but it would be a more appropriate choice.
- Minor: uncommon choice of parameters for robustness analysis. A more common choice would be $\epsilon=4/255$ via 20-step PGD or better the AutoAttack suite. This probably won't impact the observations at all but may be easier to relate to.

### Novelty
- Some claims/observations in the submitted draft were already published. This does not invalidate the (exceeding) contributions of this paper, but the respective works should at least be mentioned. [2] already performed band-masking to study the critical bands of CNNs, [1] discussed how frequencies interact with shape-bias.


### Others
- The authors use 76 pretrained models only partially citing them. The Paper Checklist clearly states to cite all the used assets.

- Authors do not discuss limitations

- Minor: The authors repeatedly claim that transformers are shape-biased. This is not inherently true. It may hold if pretrained on large datasets such as ImageNet21k, but out of the box they don't perform much better, e.g. [1] shows that XCiT just barely performs better than ResNets in shape-bias when only trained on ImageNet1k.

[1] Paul Gavrikov, Janis Keuper, Margret Keuper. "An Extended Study of Human-Like Behavior Under Adversarial Training". CVPR-W, 2023.
[2] Antonio A. Abello, Roberto Hirata, and Zhangyang Wang. "Dissecting the high-frequency bias in convolutional neural networks". CVPR-W, 2021.
[3] Remi Bernhard, Pierre-Alain Moellic, Martial Mermillod, Yannick Bourrier, Romain Cohendet, Miguel Solinas, and Marina Reyboz. "Impact of spatial frequency based constraints on adversarial robustness". IJCNN, 2021.
[4] Guillermo Ortiz-Jimenez, Apostolos Modas, Seyed-Mohsen Moosavi-Dezfooli, and Pascal Frossard. "Hold me tight! influence of discriminative features on deep network boundaries". NeurIPS, 2020.


**Questions:**

- Are there any insights on whether improving the alignment between the model critical channel compared to humans would improve robustness or shape-bias, i.e. is there evidence of causality and not only correlation?
- Did the authors attempt to encourage alignment in training?
- Are there any observations on other datasets?

**Limitations:**

The authors do not discuss limitations explicitly. One salient limitation is the evaluation of ImageNet. As discussed above certain frequency properties may not carry over to other datasets.

I don't see any potential negative societal impact of their work.

---

> ### Author Rebuttal · Authors · 2023-08-10
>
> **Strengths:**
>
> >Originality
>
> The paper explores a less common yet highly important research direction. Some "model" findings were partially reported in literature, but for the most part the findings are highly original.
>
> >Quality
>
> The submission is of high quality (minor weaknesses see below). The authors rigorously conduct (well-documented) human experiments and benchmark 76 ImageNet models to arrive at their conclusions.
>
> >Clarity
>
> >The paper is very well written. Key points are communicated clearly through writing and figures.
>
> >Significance
>
> >The authors share many findings, that may be of great interest to the NeurIPS community. In particular, they measure the human critical channel and show that models measure higher bandwidths. Adversarial Training increases the channel even further. Generally, they find correlations between channel properties and shape-bias/robustness that may lead to a new perspective on robustness but also shows another misalignment between humans and models.
>
> Thank you!
>
> **Weaknesses:**
>
> >Method
>
> >Evaluation only on ImageNet: I never thought...
>
> Thank you for raising this point. Our revised conclusion section will mention this as a limitation, as follows.
>
> *While our results are based on a large, representative sample of popular network designs, more work is required to explore the effects of diverse kinds of training data and augmentations. Additionally, our current results are purely correlational, and such experiments would also help determine if the relationship between channel properties and robustness is also causal.*
>
> >Poor choice of robust networks...
>
> >Minor: uncommon choice of parameters for robustness analysis...
>
> Thank you for these suggestions. We are working on running it with these parameters, and will try to share updates during the discussion period.
>
> >Novelty
>
> >Some claims/observations in the submitted draft were already published. This does not invalidate the (exceeding) contributions of this paper, but the respective works should at least be mentioned. [2] already performed band-masking to study the critical bands of CNNs, [1] discussed how frequencies interact with shape-bias.
>
> Done. We have written a Related work section (please see our common response to all reviewers) that will be added to the manuscript.
>
> >Others
>
> >The authors use 76 pretrained models only partially citing them. The Paper Checklist clearly states to cite all the used assets.
>
> Done. All evaluated models will be cited in the revised text.
>
> >Authors do not discuss limitations
>
> Done. We have added a discussion of limitations to the Conclusion section, as mentioned above. Copied below again for your reference.
>
> *While our results are based on a large, representative sample of popular network designs, more work is required to explore the effects of diverse kinds of training data and augmentations. Additionally, our current results are purely correlational, and such experiments would also help determine if the relationship between channel properties and robustness is also causal.*
>
> >Minor: The authors repeatedly claim that transformers are shape-biased...
>
> Thank you for pointing this out. We will change all occurrences of this statement to talk about “Imagenet-pretrained transformers)” instead of transformers in general.
>
> **Questions:**
>
> >Are there any insights on whether improving the alignment between the model critical channel compared to humans would improve robustness or shape-bias, i.e. is there evidence of causality and not only correlation?
>
> >Did the authors attempt to encourage alignment in training?
>
> Indeed, we are very interested in examining how our results are influenced by training settings, and this is something we intend to pursue next. We will modify our Discussion and Conclusion sections as follows.
>
> *DISCUSSION: Every one of the 76 networks we tested has a wide critical band. We tested all the networks from Geirhos et al., 2021 which is the largest comparison of networks and humans on object recognition. The Geirhos study included most of the popular network kinds, spanning a wide range of conditions: convolutional networks and transformers, shallow and deep networks, supervised and self-supervised training, standard and adversarially-trained networks. Thus, the conclusions of this paper are based on a representative sample of popular network designs. More work is required to explore the effects of diverse kinds of training data and augmentation.*
>
> *CONCLUSION: … are based on a representative sample of popular network designs. More work is required to explore the effects of diverse kinds of training data and augmentation.*
>
> >Are there any observations on other datasets?
>
> Not yet, but we plan to look at other datasets, as mentioned above in our modified discussion and conclusion sections (above). For now, we have considered only ImageNet since it is the most popular dataset for the evaluation of object recognition systems.

---

> > ### Comment · Reviewer_vBv1 · 2023-08-12
> > **Thank you for your efforts!**
> >
> > Thank you for your efforts! The rebuttal addresses most of my issues. Remaining issues:
> > 1. I am looking forward to the updated robustness results
> > 2. Regarding "Evaluation only on ImageNet". I am afraid that just updating the conclusion is not enough. There are multiple sections in the paper that strongly suggest that your findings scale to the detection of objects in general (e.g., L38). It would be more honest to clarify the scope of your results early on.

---

> > > ### Author Response · Authors · 2023-08-12
> > >
> > > Thank you! Yes, fair point, we'll fix these issues.

---

> > > > ### Comment · Reviewer_vBv1 · 2023-08-20
> > > > **Robustness?**
> > > >
> > > > Any there any updates on the reevaluated robustness?

---

> ### Author Response · Authors · 2023-08-21
> **Robustness re-evaluation**
>
> Thank you for the reminder and apologies for the delay in running the analysis.
>
> We re-plotted Figure 6B in the paper (Whitebox accuracy Vs Channel properties) using the adversarial attack hyperparameters that you recommended (20-step PGD with $\epsilon$=4/255). For non-adversarially trained networks, we see the same trend as before -- bandwidth is lower for networks with higher whitebox accuracy. But for adversarially trained networks, we no longer see what we saw with our previous attack parameters -- there is no significant correlation (with bonferroni correction) between whitebox accuracy and channel bandwidth.
>
> We think this absence of correlation is mainly because the new attack is much weaker than our old attack (32-step PGD with $\epsilon$=0.1), and so we see almost no difference in whitebox accuracy for networks with drastically different amounts of adversarial training. In other words, even relatively weak adversarial training is able to make networks robust to the new attack.
>
> Given that, as you said, the new attack is the more popular one, we will make sure to include this result in the supplementary material and reference it in the results section of the main paper. Thanks again for suggesting it!
>
> Also, regarding your comment about evaluation only on ImageNet, we will make sure to mention the caveat that our results apply to ImageNet and not to real-world vision, to our Introduction section, although ImageNet is often used as a proxy in existing work for natural-world-like image datasets.

---

### Official Review · Reviewer_9K3N · 2023-07-04

**Soundness:** 3 good
**Presentation:** 3 good
**Contribution:** 2 fair
**Rating:** 7
**Confidence:** 4

**Summary:**

This paper presents an analysis of neural network response to object classification from the perspective of frequency components. The authors use an analytic tool based on critical band masking, which is a technique borrowed from neuroscience.
The focus of the network to use certain frequency bands for object classification is related to the frequency capabilities of the human visual system, then with shape bias and adversarial robustness of classification networks.
The contributions/novelty is in the neuroscientific technique used to analyze the network responses and behavior. The contributions are related to comparison of the frequency bad usage in the human visual system (HSV), neural networks and adversarially-trained networks, with limited contribution in terms of practical insights or indications/suggestions on how these observations would be usable.


**Strengths:**

-	Use of a novel approach to analyze neural networks: the critical band masking technique is borrowed from neuroscience (very popular in auditory system analysis) and used to analyze the response of networks. The approach is novel and interesting.
-	Extent of models considered in the analysis (although the transformers are only slightly covered)


**Weaknesses:**

-	Usefulness and outlook of the results: while the experimental results present an interesting view on the neural networks frequency selectivity for object classification, their significance and possible usage are not clear nor explicitly covered. Networks have a different behavior/frequency selectivity (not surprisingly I would say) nut no indication of how one should take over these findings and use them.
-	Relation between HVS and neural networks: the paper reads with an implicit assumption that the human visual system and neural networks for vision tasks are comparable systems. They are actually very different (with the second showing only some conceptual resemblance with the first), in terms of processing the data, energy-efficiency, ‘training’ etc. The results show that they behave differently (as expected) but do not address substantial differences between the two systems. I see a missed answer to important questions, as ‘why should we compare these systems and what do we gain from this analysis’?
-	Details about the methodology are missing: e.g. is the analysis done taking into account both magnitude and phase of the spectrum, or only the magnitude? Phase is for instance shown important for object classification [1].
-	Placement in the literature is very limited – references are outdated: missing references of works that study the network from a frequency perspective, relating the frequency spectrum with robustness and bias problems ([2,3,4,5], to mention a few).

[1] Chen et al. Amplitude-phase recombination: Rethinking robustness of convolutional neural networks in frequency domain, ICCV 2021.

[2] Yin D. et al. A fourier perspective on model robustness in computer vision. In NeurIPS, 2019.

[3] Maiya et al. A frequency perspective of adversarial robustness, 2022.

[4] Wang et al. Frequency shortcut learning in neural networks.  NeurIPS 2022 Workshop on Distribution Shifts

[5] Wang et al. High-frequency component helps explain the generalization of convolutional neural networks. In CVPR, 2020.


**Questions:**

-	Can the authors clarify details about the method and experimental analysis, such as the use of magnitude and phase of the spectrum?
-	Can the authors clarify the motivations why neural networks based vision models have to be compared with the human visual system of the brain, and why aspects such as energy efficiency, memory, and ‘training’? are not taken into account?
-	How does the work related to actual existing work on frequency analysis of neural networks in vision tasks?
-	How the achieved results fall within the current progress in understanding neural network learning and application, and how they can be used to improve models for vision task?
-	Are 16 human observers enough to have significant results? This is glossed over very briefly in the paper, but not analyses are provided to support the claims.


**Limitations:**

The authors do not discuss limitations explicitly.
About ethics, the authors say that humans are used for experiments and that a consent form has been signed, but there is no evidence of how this form is written, what the users agree with, and who approved the experiments.

---

> ### Author Rebuttal · Authors · 2023-08-10
>
> **Strengths:**
>
> >Use of a novel approach to analyze neural networks: the critical band masking…
>
> >Extent of models considered in the analysis …
>
> Thank you!
>
> **Weaknesses:**
>
> >Usefulness and outlook of the results: …
>
> Thank you for raising this important point. We agree that the significance of the paper for engineering applications isn’t completely clear in the current manuscript. To resolve this, we will add the following text to both the abstract and conclusion of the paper.
>
> *We show that the idea of a critical-band offers a spatial-frequency-based explanation of shape bias and adversarial robustness. Thereby, our paper provides evidence suggesting that efforts to make a network more robust should look for ways to narrow its critical band.*
>
> >Relation between HVS and neural networks…
>
> *When comparing biological vision and artificial neural networks, there are two questions to be asked: what can artificial networks learn from biology, and how can neural networks inform our understanding of biology (LeCun, Bengio & Hinton, 2015; Schrimpf et al., 2018). Needless to say, the origins of deep learning relied heavily on knowledge of biology (Fukushima, 1982). But can biology impact the evolution of neural networks? Dapello et al., (2020) showed that making the first layer’s tuning match biological V1 tuning improves robustness. The critical band idea is central to understanding hearing and vision. Here we show that the critical bands, which are a measure of tuning, are different between humans and networks, and that this difference strongly explains robustness (shape bias and adversarial robustness) of networks. Thus, the results presented here are evidence that narrowing the machine critical band to match that of humans may make neural networks even more robust.*
>
> This text will be added to the discussion section in the revised manuscript.
>
> >Details about the methodology are missing: e.g. is the…
>
> Noise was generated by making an array of independent identically distributed Gaussian noise samples. This has a white spectrum and random phase. The white noise was then band-pass filtered. We did all of our experiments with random phase noise. Indeed, both machines and humans care a lot about phase while recognizing objects. We will include a sentence about this in the Discussion section.
>
> >Placement in the literature is very limited...
>
> Done. We have written a Related work section (please see our common response to all reviewers) that will be added to the manuscript.
>
> **Questions:**
>
> >Can the authors clarify details about the method and experimental analysis…
>
> Answered above in the Weaknesses section.
>
> >Can the authors clarify the motivations why neural networks …
>
> Answered above in the Weaknesses section.
>
> >How does the work related to actual existing work on...
>
> Answered above, in the newly added related work section.
>
> >and why aspects such as energy efficiency, memory, and ‘training’? are not taken into account?
>
> Thank you for raising this point. We agree that these are all interesting points that should be analyzed, and is something we intend to pursue next. The following text (see answer to next question) will be added to the discussion and conclusion sections to address these concerns.
>
> > How the achieved results fall within the current progress in understanding …
>
> Thank you for raising this important point. We agree that the significance of the paper for engineering applications isn’t completely clear in the current manuscript. To resolve this, we have added the following text to both the discussion and conclusion of the paper.
>
> *DISCUSSION: Every one of the 76 networks we tested has a wide critical band. We tested all the networks from Geirhos et al., 2021 which is the largest comparison of networks and humans on object recognition. The Geirhos study included most of the popular network kinds, spanning a wide range of conditions: convolutional networks and transformers, shallow and deep networks, supervised and self-supervised training, standard and adversarially-trained networks. Thus, the conclusions of this paper are based on a representative sample of popular network designs. More work is required to explore the effects of diverse kinds of training data and augmentation.*
>
> *CONCLUSION: … are based on a representative sample of popular network designs. More work is required to explore the effects of diverse kinds of training data and augmentation.*
>
> >Are 16 human observers enough to have significant results...
>
> We are working on analyses showing statistical significance of our human experimental results and will try to share updates during the discussion period.
>
> **Limitations:**
>
> >The authors do not discuss limitations explicitly.
>
> Done. Our revised conclusion section, copied below, will discuss limitations as follows.
>
> *While our results are based on a large, representative sample of popular network designs, more work is required to explore the effects of diverse kinds of training data and augmentations. Additionally, our current results are purely correlational, and such experiments would also help determine if the relationship between channel properties and robustness is also causal.*
>
> >About ethics, the authors say that humans are used for experiments and that a consent form has been signed, but there is no …
>
> >Flag For Ethics Review: Ethics review needed: Compliance (e.g., GDPR, copyright, license, terms of use)
>
> Clearly there has been a misunderstanding. As we point out in L128, our study and consent form were approved by the university IRB (Institutional Review Board) which abides by the Helsinki agreement (check). This is the standard protocol for review and publication of human behavior experiments in psychology and neuroscience journals. NeurIPS also follows this standard practice.
>
> It is highly unusual to request a consent form as part of a scientific review but we would be happy to share it if necessary.

---

> > ### Comment · Reviewer_9K3N · 2023-08-15
> > **Thanks for the answers.**
> >
> > Dear authors,
> > I sincerely thank you for your clear answers, especially regarding putting the significance of your work into perspective. Your rebuttal is very clear and with good argumentations. I also read the reviews made by colleagues, and your rebuttal to them.
> > I liked very much the perspective you gave about the relation between neural networks and human visual system, and how your results might be useful to improve networks in the future.
> >
> > About the ethics statement: I realize that my request made no sense wrt confidentiality of the submission. I will remove the flag.
> >
> > In summary, I appreciate the clarifications and I am willing to raise my score to Accept.

---

### Official Review · Reviewer_QrRK · 2023-07-05

**Soundness:** 3 good
**Presentation:** 3 good
**Contribution:** 2 fair
**Rating:** 4
**Confidence:** 3

**Summary:**

This work presents an empirical analysis of the sensitivity of humans and neural networks to critical band masking. This is the process of perturbing specific spatial frequencies, and determining which frequencies and at what magnitude, will lead to recognition errors. This in turns identifies the frequencies/magnitudes that a visual system “filters out” or “relies on” when performing recognition. The main finding is that neural networks are sensitive to a larger set of noise frequencies/magnitudes than human participants. Further, the paper presents evidence that adversarial-trained networks are sensitive to an even bigger set of frequencies/magnitudes, and correlation-based analyses indicating that more human-like networks that are sensitive to a narrow set of frequencies (smaller bandwidth) have a bigger shape bias.


**Strengths:**

Originality
- Critical band masking-based analysis has previously been done to analyze recognition of letters and gratings. The novelty for this paper comes from its critical band masking-based for ImageNet-like object recognition for 16 humans and 76 different neural networks.

Quality
- The design of the study is mostly sound: 1100 images from 16 categories of ImageNet with various kinds of added noise are used as the test set. Careful steps were taken to avoid clipping as a result of added noise. 16 participants are sufficient due to the small variability between subjects (Figure 2 in supplement).
- The evidence for the claims of bigger channel bandwidth than humans, inverse correlation of bandwidth and shape bias, and increased bandwidth with, are sound regarding the 76 networks evaluated in this paper. Figures 4 5 6 contain clear evidence of these claims

Clarity
- The paper is well written and mostly easy to follow. The plots in Figure 5 and 6 are quite condensed, containing a lot of information, but with the text and captions the reader can mostly parse them well.


**Weaknesses:**

Quality
- The practical motivation behind using grayscale images reduced to 20% contrast makes sense, however, this represents a testing condition outside of the training data domain of these neural networks. A control experiment where the performance discrepancy between testing on standard RGB data and contrast reduced grayscale is quantified would justify this choice of test setting.
- While the paper does investigate a diverse set of networks, one confounding factor that at least should be mentioned and ideally should be investigated in an analysis paper like this is the effect of training data and data augmentation on spatial frequency sensitivity. The design of the neural network itself is not the only factor.
    * The neural networks used in this paper are primarily trained on ImageNet, but there are other datasets, for example Office-Home, with images from different domains for each object, and therefore different spatial frequency characteristics. Training on such data will likely result in different spatial frequency sensitivity.
    * There might be straightforward steps like band-pass filtering or adding noise to the input data at training time, that might effectively reduce the range of frequencies a model is sensitive to.
- The paper lacks a related work section. While it does mention a variety of prior work throughout the draft, it would be beneficial to include a separate section that would cover prior work on topics like: spatial frequency sensitivity of humans and neural networks, shape bias, adversarial robustness, and comparing humans and neural networks more broadly. The main benefit is to help the reader understand exactly where the proposed work lies relative to prior methods.

Significance
- While it is interesting to see a comparison of spatial frequency sensitivity of networks to humans for object recognition, and to identify it as an emerging phenomenon for a variety of ImageNet-trained models, I’m concerned about what exactly is the significance of these findings for deep learning practitioners. How should researchers rethink their work going forward as a result of these findings?
- As mentioned in the quality section, after reading this paper there is a sense of it not going quite far enough in the analysis. How pervasive is the phenomenon of a wider spatial frequency sensitivity of neural networks? It would be more impactful if it is demonstrated that this is a general phenomenon despite training data or simple mitigation strategies.
- The observations about adversarial robustness are focused on one type of white-box adversarial attack, which is the least practically plausible.


**Questions:**

- What exactly is the significance of the findings presented in this work? What should be the main takeaway for researchers working on deep learning methodology for visual recognition, regarding how their future work should be affected by these findings? Is it necessarily better if the "channel" (as defined in this paper) is more human-like, and if so, why?
- Given that filtered noise perturbation significantly alters the visual appearance of an image, so much so that it appears to be quite easy to identify images with such an attack, what is the significance of identifying the susceptibility to such attacks of adversarial-trained models (described in L298)?
- What is the significance of the current findings abound frequency sensitivity, in light of the fact that the effect of training dataset choice and some basic techniques could be used to improve it?. How pervasive is this emerging phenomenon of a wider spatial frequency sensitivity of neural networks?

**Limitations:**

The authors have adequately addressed the limitations.

---

> ### Author Rebuttal · Authors · 2023-08-10
>
> **Strengths:**
>
> >Originality
>
> >Critical band masking-based …
>
> >Quality
>
> >The design of the study is mostly sound: 1100 images from …
>
> >The evidence for the claims of bigger channel bandwidth than humans, inverse correlation of bandwidth and shape bias, and …
>
> Thank you!
>
> >Clarity
>
> >The paper is well written and mostly easy to follow. The plots in Figure 5 and 6 are quite condensed, containing a lot of information, but with the text and captions the reader can mostly parse them well.
>
> Thank you!
>
> **Weaknesses:**
>
> Quality
>
> >The practical motivation behind using grayscale images reduced to 20% contrast makes sense, …
>
> Done. Fig. 2 from the attached PDF will be added to the paper’s supplementary material and will be referenced in the main paper. This figure shows the percent correct values of each network we tested for each of 3 image conditions: color, gray, and low-contrast gray. Of the 88 networks available in our source datasets, 12 were below threshold for low-contrast gray images (alexnet, squeezenet1_0, squeezenet1_1, shufflenet_v2_x0_5, bagnets 9, 17, 33, resnet50_l2_eps1,3,5, selecsls42b, selectls84), which made them impossible to test on our critical band masking task. This left us with the 76 networks that we report results for in the main paper. Importantly, for these networks, we see in the figure that the change in performance from the color to gray to low-contrast gray condition is very small.
>
> >While the paper does investigate a diverse set of networks, …
>
> Thank you for this suggestion. Yes, we totally agree that the influence of training data on these results is interesting to look at. We have clarified this point in the modified discussion and conclusion sections, as follows.
>
> *DISCUSSION: Every one of the 76 networks we tested has a wide critical band. We tested all the networks from Geirhos et al., 2021 which is the largest comparison of networks and humans on object recognition. The Geirhos study included most of the popular network kinds, spanning a wide range of conditions: convolutional networks and transformers, shallow and deep networks, supervised and self-supervised training, standard and adversarially-trained networks. Thus, the conclusions of this paper are based on a representative sample of popular network designs. More work is required to explore the effects of diverse kinds of training data and augmentation.*
>
> *CONCLUSION: … are based on a representative sample of popular network designs. More work is required to explore the effects of diverse kinds of training data and augmentation.*
>
> >The paper lacks a related work section…
>
> Done. We have written a Related work section (please see our common response to all reviewers) that will be added to the manuscript.
>
> >Significance
>
> >While it is interesting to see a comparison…
>
> Thank you for raising this important point. We agree that the significance of the paper for engineering applications isn’t completely clear in the current manuscript. To resolve this, we will add the following text to both the abstract and conclusion of the paper.
>
> *We show that the idea of a critical-band offers a spatial-frequency-based explanation of shape bias and adversarial robustness. Thereby, our paper provides evidence suggesting that efforts to make a network more robust should look for ways to narrow its critical band.*
>
> >As mentioned in the quality section, …
>
> Please refer to our answer above to your comment in the Quality section.
>
> >The observations about adversarial robustness are focused on one type of white-box adversarial attack, which is the least practically plausible.
>
> We don’t know about real-world or practically plausible attacks. Can you give us a pointer?
>
> **Questions:**
>
> >What exactly is the significance of the findings …
>
> Please refer to our answer above to your comment in the Significance section.
>
> >Given that filtered noise perturbation significantly …
>
> While the filtered noise perturbations to an image are indeed easy to see/detect, they do not affect human categorization decisions. We show in the paper that category decisions of adversarially-trained models are MORE affected by such filtered noise, making their recognition susceptible to a kind of attack that humans are not susceptible to. The significance of this is that more adversarially-robust object recognition networks are more affected by simple frequency-based perturbations which do not affect human decisions, raising questions about the robustness of these networks.
>
> >What is the significance of the current findings abound …
>
> Please refer to our answer above to your comment in the Significance section.

---

> > ### Comment · Reviewer_QrRK · 2023-08-20
> >
> > Thank you for the detailed responses to my questions. The analysis presented in the new Fig2 of the attached pdf is valuable, as it allows the reader to understand that the decrease in accuracy from color to gray and low-contrast gray does not adversely affect the findings for most models. I do not have further questions for the authors.

---

### Official Review · Reviewer_kLx7 · 2023-07-05

**Soundness:** 4 excellent
**Presentation:** 4 excellent
**Contribution:** 4 excellent
**Rating:** 10
**Confidence:** 5

**Summary:**

I have read the authors' rebuttal and will maintain my already very high rating.

This paper discovers a new fact (which should be replicated by others) about human vision: That we use the same frequency band for objects as we do for words and gratings. It seems remarkable that this has not been tested before, but a cursory google search didn’t turn up any previous result like this. Second, the paper shows that 76 deep network vision systems use a much wider band of frequencies than humans. Finally, the paper shows that aspects of these frequencies predict both shape bias and robustness against adversarial examples.

**Strengths:**

-This is one of those papers that one wishes one had written. The result about the human bandwidth is *Nature*-worthy!

-The sequence of experiments: First human psychophysics, then network “psychophysics”, then showing that these measurements of noise-sensitivity (mean, std dev., and peak noise sensitivity) account for fairly large amounts of variance in the networks’ adversarial robustness and shape-sensitivity are both logical and surprisingly revealing.

-The fact that these measurements account for 53% of the variance in shape-bias is remarkable, given that these are relatively low-level, and don’t have any necessary relationship to shape.

**Weaknesses:**

Just a few comments here:
Include the heat map key in the supplementary material (figure 2).
You should run some more experiments with adversarially-trained networks (see below).

**Questions:**

The x axis for peak noise sensitivity in Figure 6 seems incorrect, based on the definition in Figure 2F, For example, for humans, it looks to be 1/(0.03), or about 33. It’s only after checking the supplementary material that this is made clear. It’s actually 1/(0.02*2^A), where A is the fitted parameter. This should be made clearer in the main text.

Did you use MTurk’s ratings of workers? It seems like subjects 9 and 13 were only marginally engaged in the task. I would have thrown them out.

I think you should reverse Figures 1 and 2. Also (if possible) 5 & 6. In both cases, you refer to the later one before the other.

**Limitations:**

Despite using 76 different networks, they only use ResNet50 for the adversarially-trained networks. It would be good to add a couple of other networks of different types (e.g., ViT, etc.) to see that this still holds. This could be added to the supplementary material.

---

> ### Author Rebuttal · Authors · 2023-08-10
>
> **Strengths:**
>
> >-This is one of those papers that one wishes one had written. The result about the human bandwidth is Nature-worthy!
>
> >-The sequence of experiments: First human psychophysics, then network “psychophysics”, then showing that these measurements of noise-sensitivity (mean, std dev., and peak noise sensitivity) account for fairly large amounts of variance in the networks’ adversarial robustness and shape-sensitivity are both logical and surprisingly revealing.
>
> >-The fact that these measurements account for 53% of the variance in shape-bias is remarkable, given that these are relatively low-level, and don’t have any necessary relationship to shape.
>
> Thank you for rating our paper so positively! We are glad the significance and contributions of our paper came through.
>
> **Weaknesses:**
>
> >Just a few comments here: Include the heat map key in the supplementary material (figure 2).
>
> Done. The heatmap figure in supplementary material will be modified to include a colorbar and axis labels, as shown in Fig. 1 in the attached PDF.
>
> >You should run some more experiments with adversarially-trained networks (see below).
>
> Answered below.
>
> **Questions:**
>
> >The x axis for peak noise sensitivity in Figure 6 seems incorrect, based on the definition in Figure 2F, For example, for humans, it looks to be 1/(0.03), or about 33. It’s only after checking the supplementary material that this is made clear. It’s actually 1/(0.02*2^A), where A is the fitted parameter. This should be made clearer in the main text.
>
> Done. The following paragraph describing the fitting procedure will be added to Methods in the main text.
>
> *The fitting procedure is as follows. The threshold values computed earlier are first mapped to linear indices ($[ >0.16,0.16,0.08,0.04,0.02 ]\rightarrow[ 0,1,2,3,4 ]$). A Gaussian function $f(x) = Ae^{-\frac{(x-\mu)^2}{2\sigma^2}}$ having 3 parameters: peak height (A), mean ($\mu$), and standard deviation ($\sigma$) is then fit to the thresholds. These fitted parameters are converted back to their original scale and then used to calculate three properties that characterize the channel: peak noise sensitivity (reciprocal of channel height = $\frac{1}{0.02 \times 2^A}$), center frequency (frequency for peak noise sensitivity = $\mu$), and bandwidth in octaves (log full-width at half-max = $\log_2{2.355\sigma}$). An octave is a doubling of frequency. Fig. 2F illustrates the rescaled Gaussian channel (black curve) and its three parameters (orange, maroon, green) for a sample observer, along with formulae to calculate the three channel properties.*
>
> >Did you use MTurk’s ratings of workers? It seems like subjects 9 and 13 were only marginally engaged in the task. I would have thrown them out.
>
> Done. Data of subjects 9 and 13 have been discarded and all analyses involving human data have been redone and will be modified in the paper. Our results remain the same with the only significant change being that the human channel bandwidth is now narrower: 1.21 instead of the previously-reported 1.56. This makes our main result, that the network channel is wider than the human bandwidth, stronger.
>
> >I think you should reverse Figures 1 and 2. Also (if possible) 5 & 6. In both cases, you refer to the later one before the other.
>
> Done, thanks. We will remove all references to Figure 2 before those to Figure 1 because they were redundant as you pointed out. Also, the order of Figures 5 and 6 will be reversed.
>
> **Limitations:**
>
> >Despite using 76 different networks, they only use ResNet50 for the adversarially-trained networks. It would be good to add a couple of other networks of different types (e.g., ViT, etc.) to see that this still holds. This could be added to the supplementary material.
>
> Thank you for this suggestion. Yes, we agree that it will be interesting to analyze how network architecture and other factors affect our results on adversarial robustness. For this paper, we intended to test only networks that have previously been evaluated on other popular benchmarks (primarily from Geirhos et al. 2021) to observe how networks from those benchmarks fare on our spatial-frequency-based metric. Attributing our results to factors such as architecture, training data etc is interesting and is something we intend to pursue next. We have modified the Discussion and Conclusion sections of the current manuscript as follows, to include this point.
>
> *DISCUSSION: Every one of the 76 networks we tested has a wide critical band. We tested all the networks from Geirhos et al., 2021 which is the largest comparison of networks and humans on object recognition. The Geirhos study included most of the popular network kinds, spanning a wide range of conditions: convolutional networks and transformers, shallow and deep networks, supervised and self-supervised training, standard and adversarially-trained networks. Thus, the conclusions of this paper are based on a representative sample of popular network designs. More work is required to explore the effects of diverse kinds of training data and augmentation.*
>
> *CONCLUSION: … are based on a representative sample of popular network designs. More work is required to explore the effects of diverse kinds of training data and augmentation.*

---

> > ### Comment · Reviewer_kLx7 · 2023-08-18
> > **rebuttal read!**
> >
> > Sorry that your rebuttal will not cause me to raise my score! ;-)

---

### Author Rebuttal · Authors · 2023-08-10

Dear Reviewers kLx7, QrRK, 9K3N, vBv1 and TCTa

Thank you for your thoughtful reviews, with scores ranging from 4 to 10. The paper seems much improved by our efforts to respond to your questions and comments. We are particularly grateful for the explicit questions asking us how our work can be used by deep learning practitioners. Briefly, our paper provides evidence suggesting that efforts to robust-ify a network should look for ways to narrow its critical band. Most of you requested exposition of limitations and related work, which we now provide. Some of you also requested additional analyses of the statistical significance of our results, which we now provide. Reviewer TCTa had technical questions about online testing, to which we provide detailed responses addressing the concerns. We are planning to look at the influence of training and architecture on our results, similar to what reviewers suggested. Finally, as requested by two reviewers, data from two poorly performing participants was removed. Detailed responses follow.

Additionally, please find a PDF attached with helper figures. These are referenced and described in our individual responses to reviewers.

Finally, owing to lack of available space in individual responses, we post the modified Related work section as a common response to all reviewers.

**RELATED WORK**

**Spatial-frequency channels.** The visual system detects periodic patterns or gratings by means of parallel visual filters, each tuned to a band of spatial frequency (Campbell & Robson, 1968). Critical band masking  studies (Fletcher, 1940) revealed that the same single, narrow filter also mediates the recognition of letters (Solomon & Pelli, 1994; Majaj et al., 2002; Oruc & Landy, 2009), faces, and novel shapes (Oruc & Barton, 2010). Artificial neural networks also have frequency-based preferences. They are biased towards learning low-frequency functions (Rahaman et al., 2019) and prone to shortcut learning in both spatial and frequency domains (Geirhos et al., 2020; Wang et al., 2022). Existing work also suggests that robustness of a network is related to its spatial-frequency preferences (Wang et al., 2020; Li et al., 2023, Yin et al., 2019, Gavrikov et al., 2023, Abello et al., 2021).

**Shape bias.** Humans are well known to rely mainly on shape features for lexical learning and object recognition tasks (Landau, Smith & James, 1992; Geirhos et al., 2018). ImageNet-pretrained convolutional networks, on the other hand, are biased towards shape (Geirhos et al., 2018, Baker et al., 2018). Although imagenet-pretrained transformers, like humans, are shape-biased (Tuli et al., 2021), texture-vs-shape bias of networks is thought to be influenced mainly by training data and its augmentations rather than network architecture (Hermann et al., 2021).

**Adversarial robustness.** Adversarial attacks are small perturbations that cause inputs to be misclassified. (Szegedy et al., 2014, Nguyen et al., 2015). Although these perturbations are often imperceptible to humans, humans can in some cases decipher adversarial examples (Elsayed et al., 2018, Zhou & Firestone, 2019). Recent work suggests that adversarial robustness of networks relates to their spatial frequency tuning (Bernhard et al., 2021; Li et al., 2023, Maiya et al., 2021).

**Comparing human and neural network vision.** The origins of deep learning are strongly tied to neuroscience (Fukushima, 1982) and the modern convolutional network architecture was inspired by properties of the primate visual cortex (Riesenhuber & Poggio, 1999). More recently, there have been parallel efforts both to use neural networks to improve models of visual neuroscience (Yamins & DiCarlo 2016, Khaligh-Razavi & Kriegeskorte 2016, Schrimpf et al., 2018) and to improve network robustness by taking inspiration from the human visual system (Dapello et al., 2020). These lines of work strongly rely on recent advancements in model-human comparison metrics which compare networks and humans across behavior (Geirhos et al., 2021, Feather et al., 2022) and neural representations (Schrimpf et al., 2018, Kriegeskorte et al., 2008).

---

> ### Comment · Reviewer_kLx7 · 2023-08-18
> **Typo in new related work section**
>
> ImageNet-pretrained convolutional networks, on the other hand, are biased towards shape (Geirhos et al., 2018, Baker et al., 2018).
> ->
> ImageNet-pretrained convolutional networks, on the other hand, are biased towards texture (Geirhos et al., 2018, Baker et al., 2018).

---

### Decision · Program_Chairs · 2023-09-21

**Decision:**

Accept (oral)

**Comment:**

In this paper, the authors examine the properties of image recognition systems in order to better understand their divergence from the human visual system. In particular, the authors focus on a well-established procedure from psychophysics, termed critical band masking, to measure the sensitivity to spatial frequencies for humans and deep neural networks. The authors find that humans recognize objects including letters and gratings within one octave of spatial frequency. Conversely, across 76 neural networks the authors find that the networks recognize objects using a much wider bandwidth (i.e. >2x in bandwidth). The result of this observation is that neural networks are more sensitive to noise and adversarial attacks in spatial frequency regimes in which humans are insensitive.

The authors further identify that the empirically observed channel properties of neural networks may explain a good fraction of shape bias (53%) and robustness in adversarially trained networks. The reviewers commented positively on the motivation and design of the experiments, the quality and strength of the results, and the clarity of presentation. The reviewers also identified some concerns about the applicability of these results to future network design, and the lack of related work in the paper. The authors agreed to add a Related Work section and discuss more about the applicability and impact of these results on future network designs. Given that unanimous consensus of the reviews, this paper will be accepted to this conference. I took the opportunity to review this paper as well and was impressed by the overall experimental design and the potential to provide an important direction for future research in computer vision. For these reasons, I have recommended this paper for an oral presentation.